# Fitting dynamic measles models to subnational case notification data from Ethiopia: Methodological challenges and key considerations

**Alyssa N. Sbarra**[1,2]*, **Emily Haeuser**[1], **Samuel Kidane**[3], **Andargie Abate**[4], **Ayele M. Abebe**[5], **Muktar Ahmed**[6,7,8], **Tsegaye Alemayehu**[9], **Erkihun Amsalu**[10], **Aleksandr Y. Aravkin**[1], **Akeza A. Asgedom**[11], **Nebiyou Bayleyegn**[12], **Mulat Dagnew**[13], **Biniyam Demisse**[14], **Werku Etafa**[15], **Getahun Fetensa**[16], **Teferi G. Gebremeskel**[17,18], **Habtamu Geremew**[19], **Abraham T. Gizaw**[12], **Gamechu A. Hunde**[20], **Hadush N. Meles**[21], **Sibhat Migbar**[22], **Jason Q. Nguyen**[1], **Eshetu Nigussie**[23], **Rebecca E. Ramshaw**[1], **Sam Rolfe**[1], **Biniyam Sahiledengle**[24], **Noga Shalev**[1], **Yonatan Solomon**[25], **Latera Tesfaye**[26], **Gesila E. Yesera**[27], **Mark Jit**[2☉], **Jonathan F. Mosser**[1☉]

1 Institute for Health Metrics and Evaluation, University of Washington, Seattle, Washington, United States of America, 2 Department of Infectious Disease Epidemiology, London School of Hygiene & Tropical Medicine, London, United Kingdom, 3 MERQ Consultancy, Addis Ababa, Ethiopia, 4 College of Medicine and Health Sciences, Bahir Dar University, Bahir Dar, Ethiopia, 5 Asrat Weldeyes Health Science Campus, Debre Berhan University, Debre Berhan, Ethiopia, 6 Jimma University Institute of Health, Jimma, Ethiopia, 7 Cancer Epidemiology & Population Health Research Group, University of South Australia, Mawson Lakes, Australia, 8 Faculty of Health and Medical Sciences, University of Adelaide, Adelaide, Australia, 9 College of Medicine and Health Sciences, Hawassa University, Awasa, Ethiopia, 10 Sydney Medical School, Faculty of Medicine and Health, University of Sydney, Sydney, Australia, 11 Department of Environmental Health Science, College of Health Sciences, Mekelle University, Mek'ele, Ethiopia, 12 Jimma University, Jimma, Ethiopia, 13 Department of Medical Microbiology, School of Biomedical and Laboratory Sciences, College of Medicine and Health Sciences, University of Gondar, Gondar, Ethiopia, 14 College of Medicine and Health Science, Arba Minch University, Arba Minch, Ethiopia, 15 Institute of Health Sciences, Wallaga University, Nekemte, Ethiopia, 16 Wollega University, Nekemte, Ethiopia, 17 Flinders Health and Medical Research Institute, College of Medicine and Public Health, Flinders University, Adelaide, Australia, 18 Department of Reproductive Health, College of Health Sciences, Aksum University, Aksum, Ethiopia, 19 College of Health Science, Oda Bultum University, Chiro, Ethiopia, 20 School of Nursing, Faculty of Health Sciences, Jimma University Institute of Health, Jimma, Ethiopia, 21 Unit of Medical Microbiology, Department of Medical Laboratory Sciences, College of Medicine and Health Sciences, Adigrat University, Adigrat, Ethiopia, 22 College of Medicine and Health Science, Dilla University, Dilla, Ethiopia, 23 Department of Medical Laboratory Science, School of Medicine, Madda Walabu University, Robe, Ethiopia, 24 Madda Walabu University, Robe, Ethiopia, 25 Nursing Department, College of Medicine and Health Sciences, Dire Dawa University, Dire Dawa, Ethiopia, 26 Ethiopian Public Health Institute, Addis Ababa, Ethiopia, 27 Arba Minch University, Arba Minch, Ethiopia

☉ These authors contributed equally to this work.
* asbarra1@jhu.edu, jmosser@uw.edu

## Abstract

In many settings, ongoing measles transmission is maintained due to pockets of un- or under-vaccinated individuals even if the critical vaccination threshold is reached nationwide. Therefore, assessing the underlying gaps in measles susceptibility within a population is essential for vaccination programs and measles control efforts. Recently, there have been increased efforts to use geospatial and small area methods to estimate subnational measles vaccination coverage in high-burden settings, such as in Ethiopia. However, the

**Data availability statement:** This study complies with the Guidelines for Accurate and Transparent Health Estimates Reporting (see S1 Text). Case and vaccination data used in this article were obtained via a Data Use Agreement. Please contact IHME (ihme@healthdata.org) or WHO IVB (vpdata@who.int) with any data requests. All other data used are publicly available. Code used in this study can be found here: https://github.com/ihmeuw/subnational_measles_ethiopia. Estimates generated in this study can be found here: https://ghdx.healthdata.org/record/ihme-data/ethiopia-measles-incidence-susceptibility-2013-2019.

**Funding:** A.N.S., E.H., and J.F.M. received funding for this study from the Bill & Melinda Gates Foundation and Gavi, the Vaccine Alliance. A.N.S. additionally received funding from the National Institutes of Health (F31AI167535). The funders had no role in study design, data collection and analysis, decision to publish, or preparation of the manuscript.

**Competing interests:** The authors have declared that no competing interests exist.

distribution of remaining susceptible individuals, either unvaccinated or having never previously been infected, across age groups and subnational geographies is unknown. In this study, we developed a dynamic transmission model that incorporates geospatial estimates of routine measles vaccination coverage, available data on supplemental immunization activities, and reported cases to estimate measles incidence and susceptibility across time, age, and space. We use gridded population estimates and subnational estimates of routine and supplemental measles vaccination coverage. To account for mixing between age-groups, we used a synthetic contact matrix, and travel times via a friction surface were used in a modified gravity model to account for spatial movement. We explored model fitting using Ethiopia as a case study. To address data-related and statistical challenges, we investigated a range of model parameterization and possible fitting algorithms. The approach with the best performance was a model fitted to case notifications adjusted for case ascertainment by using maximum likelihood estimation with block coordinate descent. This strategy was chosen because many data observations (and likely presence of unquantified uncertainty) yielded a steep likelihood surface, which was challenging to fit using Bayesian approaches. We ran sensitivity analyses to explore variations in vaccine effectiveness and compared patterns of susceptibility across space, time, and age. Substantial heterogeneity in reported measles cases as well as susceptibility persists across ages and second-administrative units. These methods and estimates could contribute towards tailored subnational and local planning to reduce preventable measles burden. However, computational and data challenges would need to be addressed for these methods to be applied on a large scale.

## Author summary

Estimates of subnational measles susceptibility are critical for planning targeted immunization interventions. In this study, we used subnational case notifications available from Ethiopia from 2013 to 2019 as a case study to fit dynamic transmission models of measles with various reporting structures. After exploring biases inherent in these case data, we used various model fitting approaches to consider how best to include these case data in transmission models. Following our investigations, we used a deterministic optimization algorithm via block coordinate descent to fit bootstrapped models with different reporting structures (i.e., single reporting rate and region-specific reporting rates) and conducted a sensitivity analysis across multiple vaccine effectiveness values. We discussed various considerations that need to be made when fitting dynamic transmission models broadly to subnational case notification data based on their inherent biases. These include:

- accounting for sporadic temporal case reporting,

- fitting models to biased and variable case notifications despite their certainty based on statistical calculations,

- considering how best to estimate parameters that may be collinear (i.e., transmission probabilities and reporting rates),

- accounting for various reporting mechanisms and how they may contribute to under-reporting, and

- exploring implications related to assumptions on vaccine effectiveness.

## Introduction

Broadly, there is interest in conducting more targeted supplemental immunization activities (SIAs) by either age or geography in various low- and middle-income countries (LMICs), such as in Ethiopia. Recently, to contribute to informing these operations, there have been increased efforts to map geographic variation in vaccination coverage across LMICs, including Ethiopia [1–3]. Targeted interventions, such as subnational SIAs or specific routine immunization system strengthening efforts by location, might be most operationally useful in countries with moderate vaccine coverage or those with known subnational or age-group coverage heterogeneity, compared to approaches targeting by age or geography that may not be as relevant in countries with low overall coverage (where all areas or subgroups need coverage improvements via, for example, nationally-targeted SIAs). Alternatively, high coverage settings with persistent transmission among communities without explicit geographic division (e.g., among specific social networks) might need different strategies, such as strengthening of health systems and health system provisions as well as tailored communication efforts and discussions of the risks and benefits of vaccination, to increase coverage beyond the use of geographically or age-targeted campaigns.

Yet, along with other operational considerations such as resources and local factors, it would be ideal to understand who is susceptible within a population to inform targeted intervention approaches in any setting. Subnational routine measles-containing vaccine (MCV) coverage estimates, along with information on vaccine doses delivered via SIAs, afford the opportunity to begin to understand local subnational patterns of measles susceptibility. However, informing policy decisions based on vaccination coverage estimates alone would provide an incomplete picture in places where there are large amounts of natural immunity. This could yield inefficient delivery strategies if persons who were already immune due to previous infection were targeted. Additionally, serosurveys could also provide useful insight, but are costly, have biases [4], and are limited by data availability.

Nevertheless, operational decisions need to be made based on practical and logistical considerations alongside various data streams that identify communities likely to most benefit from targeted interventions to prevent avoidable measles infections in short- and medium-term time horizons. To contribute to these decisions, instead of relying on the coverage estimates alone, mathematical models incorporating transmission dynamics can potentially be used to incorporate not only information on vaccine coverage, but also demography, contact patterns, person mobility, and case notifications, to estimate the dynamics of measles within a community and quantify measles susceptibility.

However, case notifications in many LMICs, including Ethiopia, are captured via passive surveillance, and as such likely have substantial under-reporting. Given these limitations of case notifications, it is unclear to what extent this information can readily be used in models. To use case notifications to calibrate a model, it would be critical to account for case ascertainment to estimate historical measles dynamics and overall susceptibility. Reporting rates are likely to vary by location or across other factors that influence the underlying reporting mechanism or available resources.

Previous investigations have explored subnational measles susceptibility [5–13], however all have made broad assumptions that limit the interpretation of their findings. Of these previous investigations, few [6,7,9,13] attempted to quantify measles susceptibility instead of just estimating risk or relative patterns that could be used for prioritization exercises. Many of the remaining investigations [5,8,10,11] made simplifying assumptions about vaccination, such as only considering first-dose of any MCV (MCV1) coverage, neglecting to account for doses administered through campaigns, or assuming perfect vaccine effectiveness. For investigations that leveraged the use of case notifications, all but one study [10] assumed complete case

reporting, which likely greatly biases estimates of incidence and susceptibility in most settings. To quantify measles susceptibility, no published model has utilized subnational case notifications and subnational MCV1 and second-dose of any MCV (MCV2) vaccination coverage from both routine and supplemental immunization, while also systematically considering case ascertainment rates and vaccine effectiveness.

To address these methodologic gaps, research exploring new applications of compartmental modelling approaches that incorporate information from case notifications is needed. In this work, we explored methodologic considerations that would be necessary when using case notification data in transmission models to understand measles dynamics and estimate susceptibility by age, location and time, and examined limits of data quality and implied underreporting rates. We used Ethiopia as an example of a setting with these types of available case and vaccination coverage data. However, our investigations and findings are broadly applicable to other settings with similar data and epidemiologic questions related to measles susceptibility and vaccination program planning.

In 2019, Ethiopia reached 59.0% coverage [14] for MCV1 nationally, albeit with substantial subnational coverage heterogeneity [1], compared to 58.8% coverage in 2013. From 2000 to 2019, Ethiopia experienced increased national MCV1 coverage but also increases in geographic inequality, suggesting emerging specific geographies with increased relative vulnerability to ongoing measles transmission. Ethiopia introduced MCV2 into the national immunization schedule in 2019 [15]. Within the first year of introduction, routine MCV2 coverage reached 43.4% nationally [14]. In 2017, Ethiopia executed two subnationally-targeted measles campaigns (Table B in S1 Text), or SIAs, in March and August which combined delivered over 23,750,000 vaccine doses [15] (with the 2017 total population size of approximately 100,000,000, and approximately 15,000,000 children under age five [16]). Likely due to low coverage and geographic heterogeneities, in 2019, Ethiopia reported almost 4,000 measles cases nationally [17]. We investigated subnational case data along with estimates of vaccination coverage to explore various modelling challenges and considerations.

## Methods

### Case notifications

We obtained subnational case notifications of suspected measles from across Ethiopia from 2013 (the first year that they were available) to 2019 in 5-year age bins (e.g., 0–4-year-olds, 5–9-year-olds, etc.) that were collected through routine surveillance systems by epiweek (i.e., ISO week shifted one day to begin on Sunday). In Ethiopia, there are 11 regions (i.e., first administrative units), 79 zones (i.e., second-administrative units), and over 700 districts (i.e., third administrative units). For each subnational location reported within the case data, we identified the most appropriate corresponding zone based on the Database of Global Administrative Areas administrative boundaries [18]. We compared the distribution of reported incidence by region and age bin. We additionally aggregated subnational, age-specific case notifications nationally and across all ages to compute a suggested national reported incidence and compared to the actual national, age-aggregated reported incidence. We also compared MCV1 coverage across zones by year to the reported number of cases across all ages by year in each zone and computed their corresponding correlation. To examine stochasticity in reporting, we assessed the number of zones reporting 0, less than 10, and less than 100 cases across available years and additionally computed the number of times zones reported cases with a single week gap.

We used MCV1 and MCV2 vaccine coverage estimates from either routine immunization (RI) or SIAs. These estimates rely on spatial estimates of MCV1 and MCV2 coverage,

as well as a computed metric of campaign efficiency used to allocate doses of administered during campaigns to either previously vaccinated or unvaccinated children. These coverage estimates are age-, epiweek-, and zone-specific and are generated using a cohorting model to track coverage across age groups over time and space. These estimates are based on previously published work [1] and additional details can be found in Section 1 in S1 Text. In short, household-based survey data on MCV1 and MCV2 coverage are geolocated to the most geographically granular level possible. For RI MCV1 coverage, we used a model-based geostatistical framework to estimate coverage under the assumption that coverage is more similar in years and locations that have similar covariate patterns and in years, locations and age groups that are closer across time, space and age. For RI MCV2 coverage, less data was available to include in models, as MCV2 was only introduced in Ethiopia in 2019. Therefore, we used a hierarchical model at the zone level. For combined RI and SIA coverage, we developed a model tracking cohorts through time, accounting for RI coverage increases along with doses reported to be administered during SIAs.

## Dynamic model structure

An overview of our modelling structure can be found in Fig 1 and an overview of main parameters in Table C in S1 Text. We then used these subnational case notifications to fit a dynamic, zone-level, transmission model across weeks from 2013 to 2019 in 24 age groups, with smaller intervals for ages with high measles incidence (12 for monthly bins for 0–11-month-olds, 4 for yearly bins for 1–4-year-olds, 1 bin for 5–9-year-olds, 1 bin for 10–14-year-olds, and 5 bins 10 years wide for 15–64-year-olds, and 1 bin for 65-year-olds and

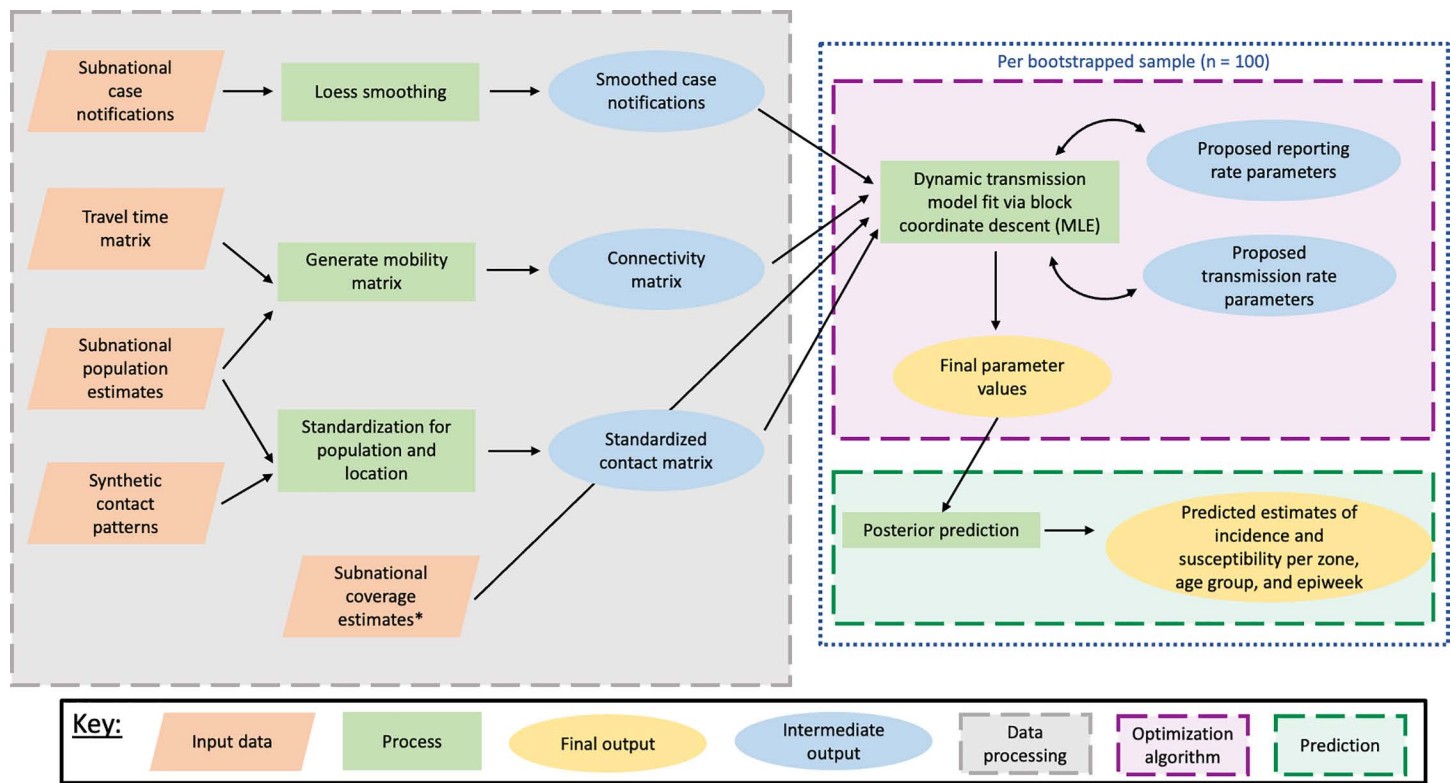

**Fig 1. Transmission modelling functional block diagram.**

older). We used demographic information from WorldPop gridded population surfaces [19] calibrated to population sizes from the Global Burden of Disease study [16,20] as age-specific population and live birth counts by zone. We linearly interpolated annual population sizes to epiweeks and assumed a constant weekly birth rate for each year. For each zone, we used a synthetic contact matrix [21] standardized for zone population within each epiweek.

To estimate transmission between zones, we used a gridded friction surface of travel time by motorized vehicle [22] to compute the travel time in minutes between every combination of population-weighted centroids from all zones. We used these values to construct a modified gravity matrix ($G$) for each pair of zones ($z$ and $y$), such that:

$$G_{z,y} = \frac{P_z P_y}{V_{z,y}},$$

where $P_z$ and $P_y$ are population sizes and $V_{z,y}$ is the distance in minutes computed from the friction surface. We used our modified gravity matrix [23] $G$ to compute a mobility matrix ($K$), such that:

$$K_{z,y} = \frac{G_{z,y}}{G_Y} * (1 - \psi), \text{when } z \neq y$$

$$K_{z,y} = \psi, \text{when } z = y,$$

where $G_Y$ are the sums of the columns of $G$ and $\psi$ is the probability of persons staying in their home zone in a given epiweek. To aid in model identifiability, we assume $\psi$ to be 0.99 based on in-country expertise.

We used a time-varying compartmental model to track the proportion of persons in each age group and zone that were maternally immune, susceptible, infected, and recovered across epiweeks from 1980 to 2019. For each compartment, we maintained information on the proportion of persons who were unvaccinated, vaccinated with 1 dose of MCV, and those who were vaccinated with 2 or more doses of MCV (Fig 2). We defined a constant starting state based on assumptions of population-level immunity in a pre-vaccine era [24,25], such that 25% of infants under 6-months-old, 60% of 6-to-8-months, and 98% of all other persons were recovered. Additionally, we assumed that 40% of infants aged under 6-months and 10% of infants aged 6-to-8-months were maternally immune. All other persons were considered susceptible at our starting state apart from 4253 infected individuals to seed the infection. This was computed empirically by estimating infections in the first epiweek by taking the national, annual reported cases from 1980, dividing by number of epiweeks and applying an approximately 5% reporting rate. We distributed initial cases across age groups based on the age pattern observed from the earliest age-specific case notification data available (i.e., from 2013). We tested the sensitivity of these assumptions, and they yielded little to no difference in results (i.e., a difference of fewer than 100 estimated cases in the final year across zones), likely as the model had ample time to equilibrate since it was run for 33 years (i.e., 1749 epiweeks) before fitting to the first available data in 2013.

For each time step or epiweek ($w$), we computed the transmission probability, also known as the flow from the susceptible to infected compartments, ($\beta_w$) such that:

$$\beta_w = A * sin\left(2\pi * \frac{w}{53} + 2\right) + D$$

$$D = \frac{\beta_{max} + \beta_{min}}{2}$$

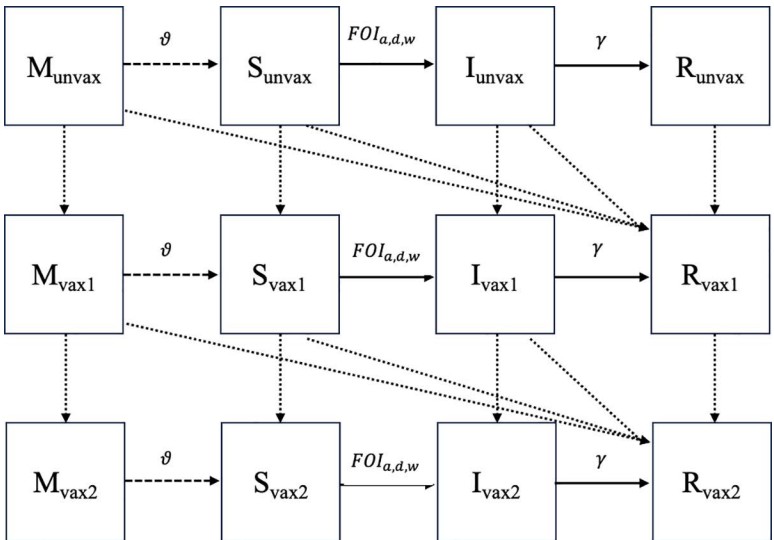

**Fig 2. Transmission modelling flowchart.** Compartmental flowchart of transmission model. M represents the maternally immune class, S represents the susceptible class, I the infected class, and R the recovered class. Solid lines represent transitions following infection, dashed lines represent transitions following loss of maternal immunity, and dotted lines represent transitions following either successful or unsuccessful vaccination events. Unvax compartments are with unvaccinated persons, vax1 vaccinated with 1 dose of MCV, and vax2 with 2 or more doses. $\vartheta$ represents the rate of maternal immunity waning, $FOI_{a,d,w}$ represents the force of infection for age group $a$ in zone $d$ in epiweek $w$, and $\gamma$ represents the recovery rate. Demographic and vaccination transitions were omitted from this diagram for simplicity.

$$A = \frac{\beta_{max} - \beta_{min}}{2}$$

$$\beta_{max} > \beta_{min}$$

where $\beta_{max}$ is the maximum transmission probability bounded between 0.1 and 1, $\beta_{min}$ is the minimum transmission probability bounded between 0 and 1, $A$ is the amplitude, and $D$ is the vertical displacement. The seasonal displacement was calculated empirically by fitting a sine curve to reported case notifications and was determined to be approximately 2 (i.e., at the 13 week of each year). We then computed the force of infection ($FOI$) for each time step or epiweek ($w$), zone ($d$), and age group ($a$) using the following equation:

$$FOI_{a,d,w} = \beta_w * \sum_{c=1}^{C}\sum_{z=1}^{Z}\left(W_{a,c,w,d} * K_{z,d} * I_{w-1,c,z}^{\alpha}\right),$$

where $W_{a,c,w,d}$ is the contact rate from our synthetic contact matrix between age groups $a$ and $c$ standardized to the population size in time $w$ and zone $d$, $K_{z,d}$ is the proportion of persons from zone $z$ traveling to zone $d$ in a given time step, and $I_{w-1,c,z}$ is the proportion of persons in age group $c$ from zone $z$ who were infected in the previous time step. $\alpha$ is a parameter, assumed to be 0.99 for purposes of model identifiability, to account for mixing parameters of the contact process or the discretization of a continuous process [26,27].

In each time step, we used the $FOI$ to estimate the newly infected persons in each age group and zone. We assumed the recovery rate ($\gamma$) serial interval to be 2 epiweeks and that maternal immunity wanes exponentially ($\vartheta$) starting at 4-months-old [28]. Finally, based on

already discussed RI and SIA MCV1 and MCV2 coverage values, we re-calibrated to weekly population-level vaccination prevalence.

We tested several model fitting algorithms, including Markov chain Monte Carlo (MCMC) samplers and block coordinate descent (described in detail below). For each, we estimated the following parameters: maximum and minimum transmission parameters over a season ($\beta_{max}$ and $\beta_{min}$) and a reporting rate ($\rho$). We assumed the following distribution of cases:

$$C_{a,d,w} \sim NegativeBinomial\left(I_{a,d,w} * \rho, 5\right),$$

where $C_{a,d,w}$ is the reported number of cases in zone $d$, age group $a$ and time step $w$, $I_{a,d,w}$ is the down-adjusted number of estimated cases in zone $d$, age group $a$ and time step $w$, and $\rho$ is a reporting rate. We tested various dispersion parameters (i.e., 1, 5, 10, 50) to improve likelihood acceptance in early modelling experiments. Increasing the dispersion parameter greater than 5 did not improve initial sampler acceptance rates, so therefore we used 5 for the remainder of experiments. We used R version 4.2.2 [29] for this analysis and the *Rcpp* package version 1.0.9 [30] to build our transmission model. We ran all computations in a distributed computing environment with 50GB memory and 4 cores. We explored data-related considerations including sporadic case reporting, vaccine effectiveness, and variation in reporting rates. Additionally, we considered the implications of steep likelihood surfaces and collinear parameters before selecting a final model fitting algorithm. All data processing, final model, and diagnostic code can be found here: https://github.com/ihmeuw/subnational_measles_ethiopia.

## Results

### Case notifications

Cases were reported across zones and regions in Ethiopia from 2013 to 2019 with varying seasonal patterns and annual magnitudes (Fig 3). Twenty-one (of 79) zones reported fewer than 100 cases across the entire seven-year (371-epiweek) period, nine of which reported fewer than ten cases, and six of which reported no cases (Fig D in S1 Text). Per zone, on average, a one-week gap (i.e., interval with zero cases) in reported cases occurred 46 times (SD = 27.2). One zone reported cases with a one-week gap 105 times. There were no zones that reported at least one case every week. The sum of available cases reported subnationally were aggregated to annual, national, all-age values of reported cases to compare to cases reported via the Joint Reporting Form. Temporal patterns of cases were not consistent between both sources (Fig 4), with the aggregated subnational cases having relatively little temporal variation nor matching the large outbreak reported within national case notifications in 2015. It was unclear whether case data from subnational locations or national cases were more accurate.

The age distribution of reported cases across most years stayed consistent over years and yielded approximately half (55%) the number of cases reported among 5-to-9-year-olds than among 0-to-4-year-olds (Fig 5). The number of reported cases among 10-to-14-year-olds was approximately two-thirds (66%) of the number reported among 5-to-9-year-olds. The observed age pattern of reported cases is unexpectedly that typically seen from countries or locations approaching measles elimination [31], not from locations with endemic transmission with relatively moderate vaccine coverage (such as Ethiopia). Additionally, we aggregated cases across zones by year and compared to MCV1 coverage from RI or SIA among 0-to-4-year-olds (Fig 6). Reported case notifications by zone-year are not correlated with coverage among 0-to-4-year-olds in the same zone year (Pearson's product-moment correlation, t = -1.705, p = 0.089). The lack of negative correlation between reported incidence and vaccine

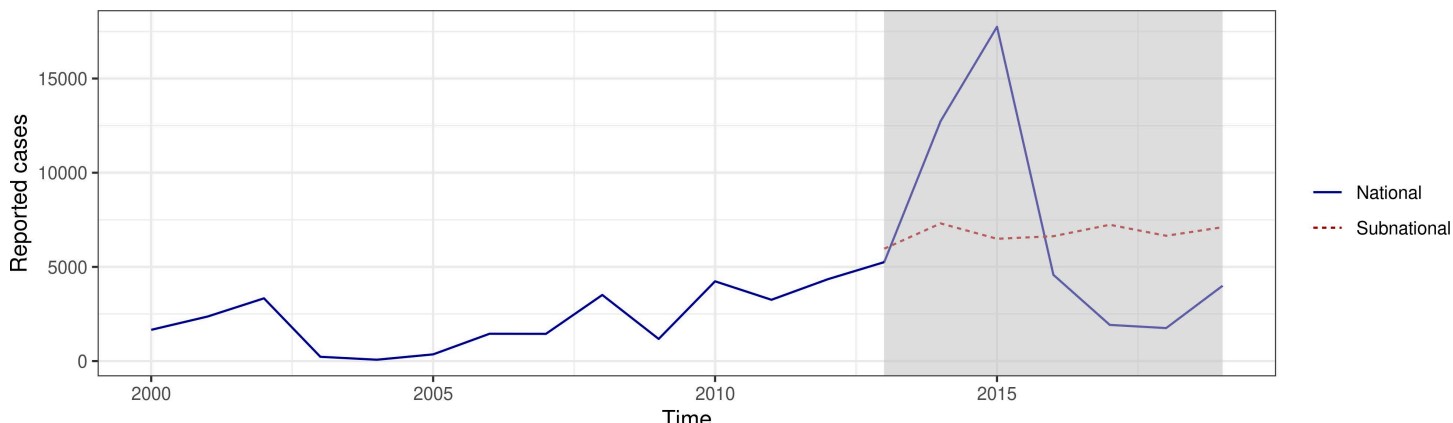

**Fig 3. Raw reported weekly suspected measles case notifications nationally and by region (i.e., first-administrative unit) in Ethiopia from 2013 to 2019.** Across regions, different y-axes are used to emphasize the underlying spatial and relative temporal pattern within each region.

**Fig 4. Reported annual suspected measles case notifications nationally (blue, solid line) from 2000 to 2019 via Joint Reporting Form and aggregated subnational case notifications to national scale (red, dotted line) from 2013 to 2019 during the study period (grey).**

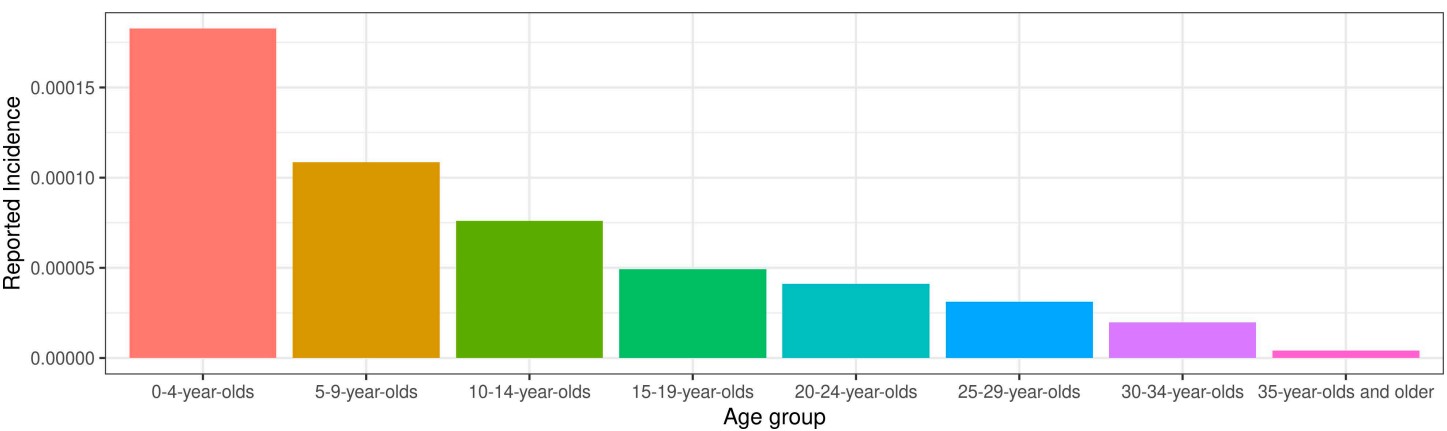

**Fig 5. Reported measles incidence in Ethiopia in 2019 across reported five-year age groups.** Reported measles incidence from Ethiopia was generated from suspected measles cases.

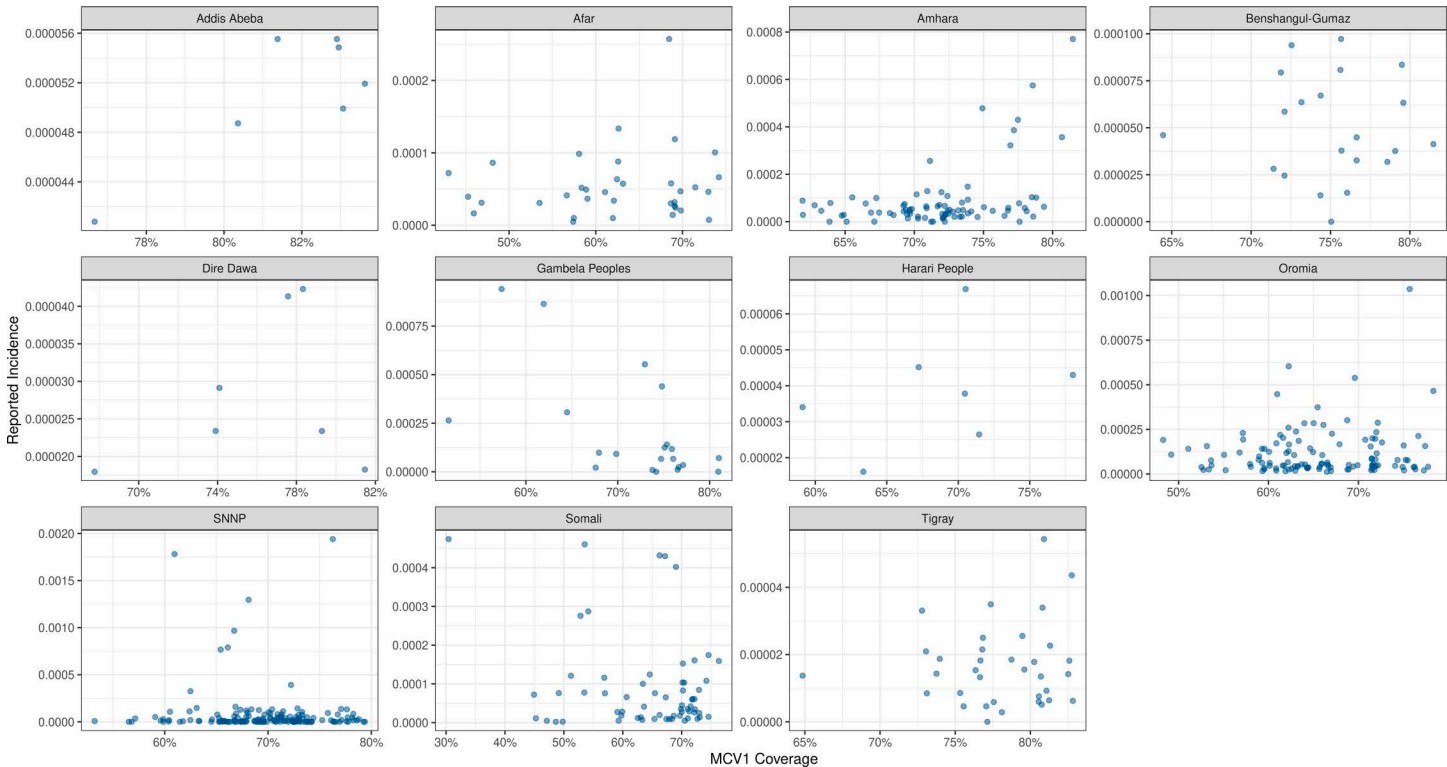

**Fig 6. Reported measles incidence across regions in Ethiopia against first-dose measles-containing vaccine (MCV1) coverage for each zone-year among 0-to-4-year-olds from 2013 to 2019.** Reported measles incidence rate from Ethiopia was generated from suspected measles cases. Across regions, different y-axes are used to emphasize the underlying reported incidence rate within each region.

coverage suggests variability in case reporting, inaccuracies in estimates of vaccine coverage (such as errors in the numerator and/or denominator), variability in vaccine effectiveness, or some combination of these factors.

## Data considerations for model fitting: Sporadic case reporting

As observed at the regional level (Fig 3, in Benshangul-Gumaz, Dire Dawa and Harari People regions), many zones reported cases only every other week, which is inconsistent with epidemiological expectations for measles transmission and may represent a data artefact generated in the reporting process. However, it was difficult to tell from the data whether the zones that reported alternating weeks with zero cases represent the total sum of cases across two or more weeks or whether these are missing other values that were not reported. We hypothesized it was most likely that cases were being aggregated across weeks prior to reporting, such that the alternating weeks were the sum of cases from every two or more weeks.

We explored a range of methods to account for noise in base notification data before their inclusion in model fitting. We first explored aggregating cases by month or annual quarter. However, this approach was highly sensitive to small differences in the timing of peak transmission rates. In some cases, the model, which is run at the epiweek level, predicted a marginally different peak transmission time than the observed data (Fig A in S1 Text). If these predicted transmission peaks fell in a separate month or quarter from the data, however, trivial differences in timing would result in poor model fits using likelihood-based fitting methods when all results were aggregated to the coarser time step. Conversely, larger differences in transmission peaks were not penalized, as long as the modeled and observed peaks fell in the same quarter.

Despite likely small differences in specific weekly timing between major transmission events, in these instances, likelihood calculations suggested poor model fits. Instead, we chose to smooth cases by zone and age group using a locally estimated scatterplot smoothing (LOESS) function with a span of 0.2. We tested LOESS functions with multiple spans for sensitivity, but ultimately chose 0.2 to maintain underlying variation across the time series and avoid over-smoothing. This process yielded a smooth time series of cases in each zone and age group, which allowed for the model to fit to the average of cases across a time period and mitigate fitting challenges stemming from the noisy time series.

## Data considerations for model fitting: Vaccine effectiveness

We initially assumed that vaccine efficacy was 93% for each dose given [32] (for either first- or second-dose, i.e., the second dose seroconverts 93% of all seronegatives who receive it, including vaccine failures from the first dose) and independent of the number of doses previously received. However, this assumption yielded little to no transmission (Fig B in S1 Text) in more recent years (i.e., after 2008) even with complete reporting due to high recorded vaccine coverage, which are both implausible as they do not match recent case reports. Therefore, we added a parameter ($vax\_eff$) in our model to represent vaccine effectiveness, which ultimately increased the proportions of susceptible individuals in our dynamic system to foster transmission events occurring in later years. This vaccine effectiveness term was capped at 93% to align with previous estimates of optimal measles vaccine efficacy [32].

## Data considerations for model fitting: Variation in reporting rates

Based on observed trends in the subnational case data from Ethiopia (e.g., discordant trends in reported cases nationally and subnationally observed in Fig 4), we suspected that there were likely differential rates of case ascertainment by geography. To further explore structures for reporting rates, we tested the implications of applying different reporting rates during model

fitting. We assumed that cases followed a negative binomial distribution and fit cases to incidence adjusted for reporting rates ($\rho_{a,d,w}$) from our modelling output. We first tested a single reporting rate $\rho$ across all age groups, regions, and years, such that:

$$logit\left(\rho_{a,d,w}\right) = logit\left(\rho\right)$$

To explore variations in reporting, we tested a model with region-specific reporting rates, $\rho_Q$, for each region $Q$, that is used across all zones $d$ in region $Q$ such that:

$$logit\left(\rho_{a,d,w}\right) = logit\left(\rho_Q\right)$$

## Statistical considerations for model fitting: Steep likelihood surface

We fit a high-dimensional model with over 260,000 likelihood contributions from data observations that overall have unmeasured biases and uncertainty. Despite these data limitations, many observations yielded a very steep likelihood surface which suggested little data uncertainty (regardless of underlying statistical distribution assumed), and as such Bayesian methods using any kind of sampler had a very challenging time accepting proposed samples. For illustration, if the input data were perfectly replicated in our model results, our model would yield a log-likelihood value of -57603 (i.e., the highest possible log-likelihood, given the sheer volume of likelihood contributions). Adding just 0.1 case to each observation – a trivial difference, in practical terms – would yield a log-likelihood of -75607, or a log-space difference of -18004, which is far too great to be accepted by MCMC samplers. This example demonstrates the steepness of the likelihood surface, which limits the ability of most conventional MCMC approaches to explore the surface appropriately and provide a reasonable quantification of uncertainty. We tested various algorithms for model fitting including the following: MCMC, MCMC with parallel tempering, adaptive MCMC, rejection sampling, rejection sampling of Sobol hypercube samples, and a deterministic optimization algorithm.

In subsequent models using MCMC, samplers had very limited ability to explore the full parameter surface and often accepted very few samples. In attempt to combat these issues, we explored alternative forms of MCMC including implementing MCMC with parallel tempering [33] (Fig C in S1 Text) and adaptive MCMC samplers [34]. Ultimately, neither alternative approach substantially accepted more samples and both approaches failed to overcome the fundamental steepness of this surface. Additionally, because we were fitting a high-dimensional model with a relatively slow likelihood calculation (approximately 20 seconds), allowing the model to run over many hundreds of thousands of iterations in hopes of eventual convergence became computationally expensive and unfeasible.

We then explored generating proposed samples without using a Bayesian framework; we did this by externally generating parameter samples via a Sobol hypercube to select a quasi-random set of parameter values to use in a rejection sampler. However, the root of the problem (i.e., steep likelihood surface) remained, with very few samples being accepted (i.e., various runs yielded between 1 and 6 samples out of 10000 selected).

As such, we addressed this issue using a deterministic optimization of a derivative-free maximum likelihood estimator (via the *dfoptim* package version v2020.10-1 [35]). Models were able to successfully run while using a fraction of the computational resources (e.g., convergence in 5 hours relative to MCMC runs that failed to converge after 72 hours).

## Statistical considerations for model fitting: Collinear parameters

Transmission parameters, vaccine effectiveness, and reporting rates are not independent from one another when estimating the underlying dynamics of measles within a community. For

example, given constant transmission parameters, for a given number of reported cases, lower vaccine effectiveness typically corresponds to lower reporting rates, as higher true incidence rates are expected.

Given the collinearities among our parameters being estimated (i.e., transmission parameters, vaccine effectiveness, and reporting rate), we needed to make additional modifications to our model fitting algorithm. The first was to remove vaccine effectiveness as a parameter that was directly estimated in our modelling framework. Instead, we decided to use a profile likelihood two-stage estimation approach across different vaccine effectiveness parameters to select the model with the best fit as determined by statistical criteria (i.e., Akaike Information Criterion [AIC] score).

This left us with transmission parameters and reporting rate to fit, which are still collinear (i.e., low transmission rates suggest a high reporting rate as less transmission would yield overall lower numbers of predicted cases, and conversely, high transmission rates suggest a low reporting rate as more transmission would yield higher numbers of predicted cases). When fitting models that estimated both transmission rates and reporting rates, we additionally noted sensitivities to the starting state when using a single deterministic maximum likelihood estimation (MLE) algorithm (Tables D1 and D2 in S1 Text). As an alternative to fitting a single model to estimate both sets of parameters, we instead used block coordinate descent. In this approach, we first fit a reporting rate using deterministic optimization via the *stats* package version 4.2.2 [36] given an initial starting state of transmission parameters in an accepted range of $R_0$ values for measles (i.e., 9 – 19; the average number of cases arising from a single infected individual in a fully susceptible population) [37–39]. Initial values parameters were as follows: $\beta_{max}$ = 0.25, $\beta_{min}$ = 0.12, and all reporting rates $\rho$ and $\rho_Q$ = invlogit(-5).

We used package default convergence and optimization settings. Then we fit another deterministic optimization via the *dfoptim* package version v2020.10-1 [35] to estimate the transmission parameters using the fitted values for a reporting rate estimated from our first step. We used default convergence and optimization settings except allowing 20,000 maximum objective functions to be evaluated.

## Final model fitting algorithm

We ultimately used MLE via block coordinate descent to first fit our reporting rate parameters (i.e., either $\rho$, or $\rho_R$ by region depending on the reporting structure for the model) while holding $\beta_{max}$ and $\beta_{min}$ constant, and then subsequently to fit $\beta_{max}$ and $\beta_{min}$ while holding our reporting rate parameter value(s) constant. This sequence (i.e., fit reporting rate parameter(s), then transmission parameters) was repeated iteratively 10 times, as typically by iteration 6–7 the algorithm yielded negligible changes (i.e., less than 0.001) in parameter values. We selected initial values for $\beta_{max}$ and $\beta_{min}$ of 0.25 and 0.12 respectively based off a plausible range of corresponding $R_0$ values (i.e., 9 – 19).

To generate uncertainty, we fit 100 bootstrapped samples of parameter values and likelihoods. Zone-weeks were selected for inclusion randomly such that 25% of zone-weeks were included per bootstrapped sample (among all age groups). Across various reporting structures and among a sensitivity analysis of vaccine effectiveness values, we selected the model with the lowest AIC score based on the median likelihood value across bootstraps. Across bootstraps, each block coordinate descent algorithm was run for 10 full iterations, regardless of if convergence had been met prior to this interval. Generally, convergence was reached before approximately 4–5 iterations (Fig F in S1 Text) as evidenced by unchanging parameter results with further optimization. We tested four values of overall vaccine effectiveness (47%, 70%, 82%, and 88%) to account for both estimated 93% vaccine efficacy from prior studies, as well as faults in cold chain or other reasons biologically associated with lack of seroconversion (e.g.,

malnutrition). We used our posterior bootstrapped samples to compare regional reporting rates to regional metrics of socio-demographic index [16], which is a composite metric of fertility, education and lag-distributed income per capita, and to make predictions of measles incidence and susceptibility across age groups, zones, and weeks from 2013 to 2019. We calculated $R_0$ using a next generation matrix approach (Fig E in S1 Text) such that for each zone $d$ across age groups $a$ and $c$, we computed the following:

$$\delta_{a,c} = \beta \star W_{a,c,2,d}$$

$$\tau_d = \max\left(eigen(\delta)\right),$$

such that $\beta$ is a transmission probability and $\tau_d$ is the zone-specific $R_0$ value. We took the mean $\tau_d$ across all zones to calculate the expected $R_0$ from each possible $\beta$ value.

## Model fitting: Results

We fit bootstrapped samples using block coordinate descent across two different reporting structures (i.e., a single reporting rate and region-specific reporting rates) and four different vaccine effectiveness values; see Section 2 in S1 Text for full model results and the best model results per reporting structure including Tables E and F in S1 Text for model performance metrics across reporting structures. The model with the lowest AIC score was with both region-specific reporting and a vaccine effectiveness of approximately 47%. In this model, the maximum and minimum transmission parameters were 0.135 (95% uncertainty interval (UI): 0.134 – 0.137) and 0.102 (95% UI: 0.102 – 0.102) respectively, which corresponds to a range of $R_0$ values from 7.5 to 9.8 (Fig E in S1 Text). Block coordinate descent iterations for all parameter values are in Fig F in S1 Text. Estimated reporting rates ranged from 0.26% (95% UI: 0.24 – 0.27%) in Tigray to 2.93% (95% UI: 2.53 – 3.35%) in Gambela region. Regional reporting rate was not associated with socio-demographic index (Pearson's product-moment correlation, t = 0.856, p = 0.415). Incidence and case predictions across weeks and zones by age groups (0-to-4-year-olds, 5-to-9-year-olds, and 10-to-14-year-olds) are available in Figs 7 and G–K in S1 Text.

## Subnational susceptibility patterns

We compared subnational patterns of susceptibility (i.e., among persons without immunity from vaccination, natural infection, or maternal immunity) in 2019 suggested from the median of our modeled, bootstrapped results from both models with a single national reporting rate and regional reporting rates. Among 0-to-4-year-olds, susceptibility results from models with regional reporting rates were correlated with coverage estimates (Figs 8 and 9, among 0-to-4-year-olds via Pearson's product-moment correlation p < 0.001). When using a model with a single reporting rate compared to a regional reporting rate, there were negligible differences in susceptibility patterns across 0-to-4-year-olds (i.e., less than absolute differences of 1% in proportions), as approximately the same transmission parameters were estimated.

Estimated susceptibility varied across ages by zone. Temporal trends in age susceptibility also vary across zones (Fig 10), likely largely owing to the recency of mass campaign events. Across years with available data, we estimated a higher proportion of 1-to-4-year-olds susceptible in 2019 compared to 2013 across regions (Fig 11). Susceptibility trends were observed to be more consistent across 0-year-olds between 2013 and 2019, likely largely as a result of consistent assumptions of maternal immunity across the study period. In 2019 across 0-to-4-year-olds, zones in Somali region (i.e., the region bordering Somalia, which has recently

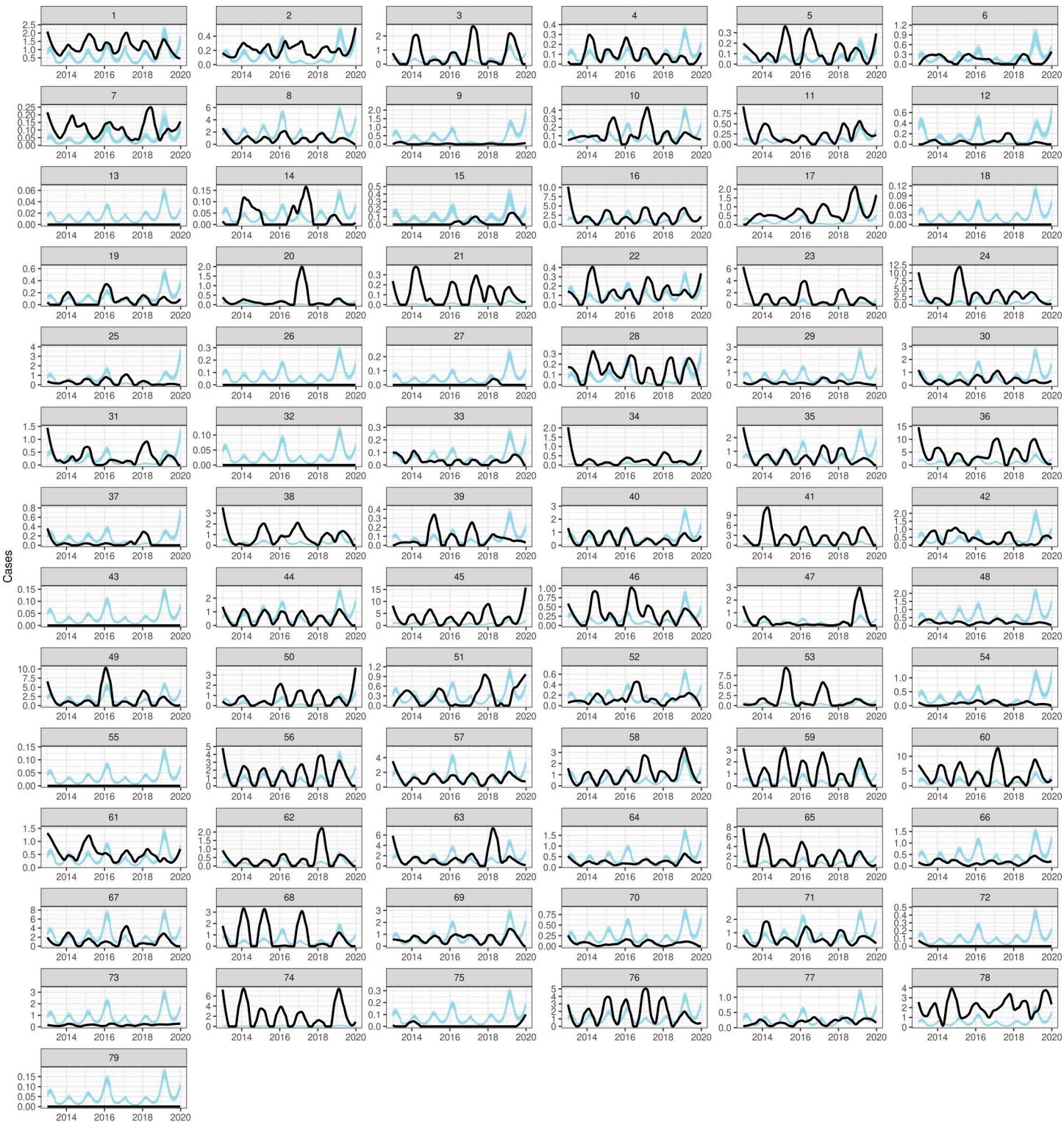

**Fig 7. Smoothed reported suspected measles incidence among 0-to-4-year-olds (light blue) compared to estimated incidence adjusted for reporting (black) from best model fit across each zone in weeks from 2013 to 2019.**

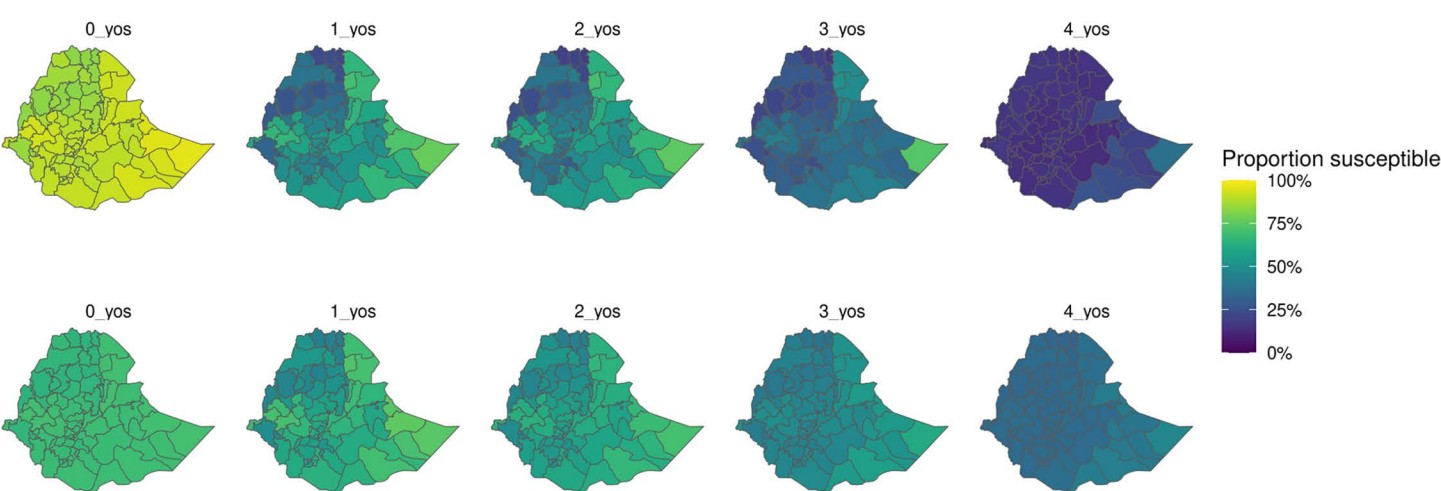

**Fig 8. Proportion susceptible in 2019 in Ethiopia among 0-to-4-year-olds, top panel unvaccinated (i.e., 1 – MCV1 coverage from routine or supplemental immunization) and bottom panel from modelled outputs (i.e., not maternally immune, immunity from vaccination, or immunity from previous infection).** Base layers used in maps were obtained from the Global Database of Administrative Units (https://gadm.org/).

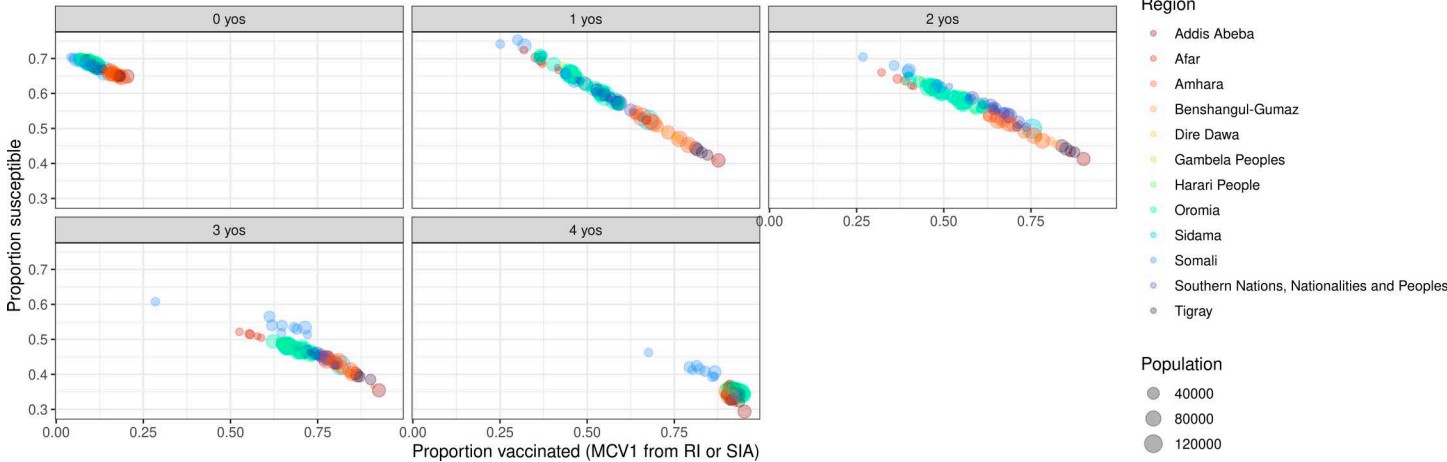

**Fig 9. Proportion susceptible across each zone in 2019 in Ethiopia among 0-to-4-year-olds among unvaccinated (i.e., 1 – MCV1 coverage from routine or supplemental immunization) and modelled outputs (i.e., not maternally immune, immunity from vaccination, or immunity from previous infection).** Points for each zone are colored by region and sized by population.

experienced political instability) had the highest susceptible proportions compared to zones from other regions.

## Discussion

Considering subnational measles dynamics and heterogeneity is critical for identifying areas that, along with other operational factors, may benefit from targeted interventions to address local (i.e., zone-level) susceptibility that are responsible for driving ongoing measles transmission. However, there is currently no "gold-standard" data set available to fit models that reflect these heterogeneities. Seroprevalence data are sparse and present biases related to test sensitivity, as well as challenges in distinguishing between seropositivity due to

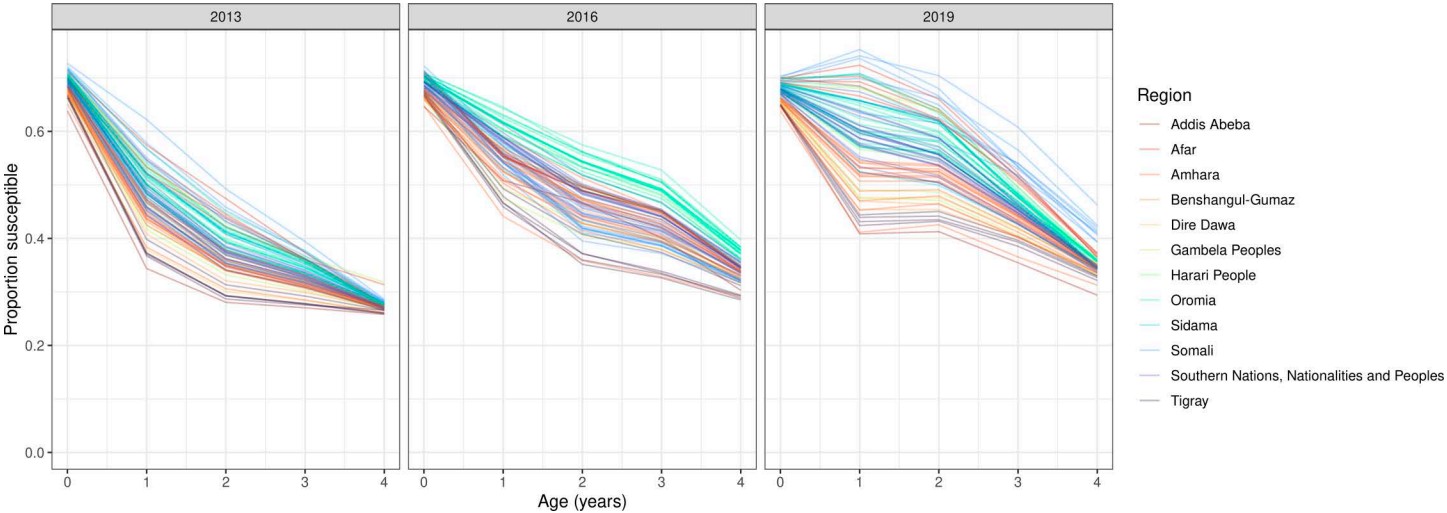

**Fig 10. Susceptibility by age (among 0-to-4-year-olds) by zone and year from years 2013, 2016 and 2019.** Each line represents a zone and is colored by region.

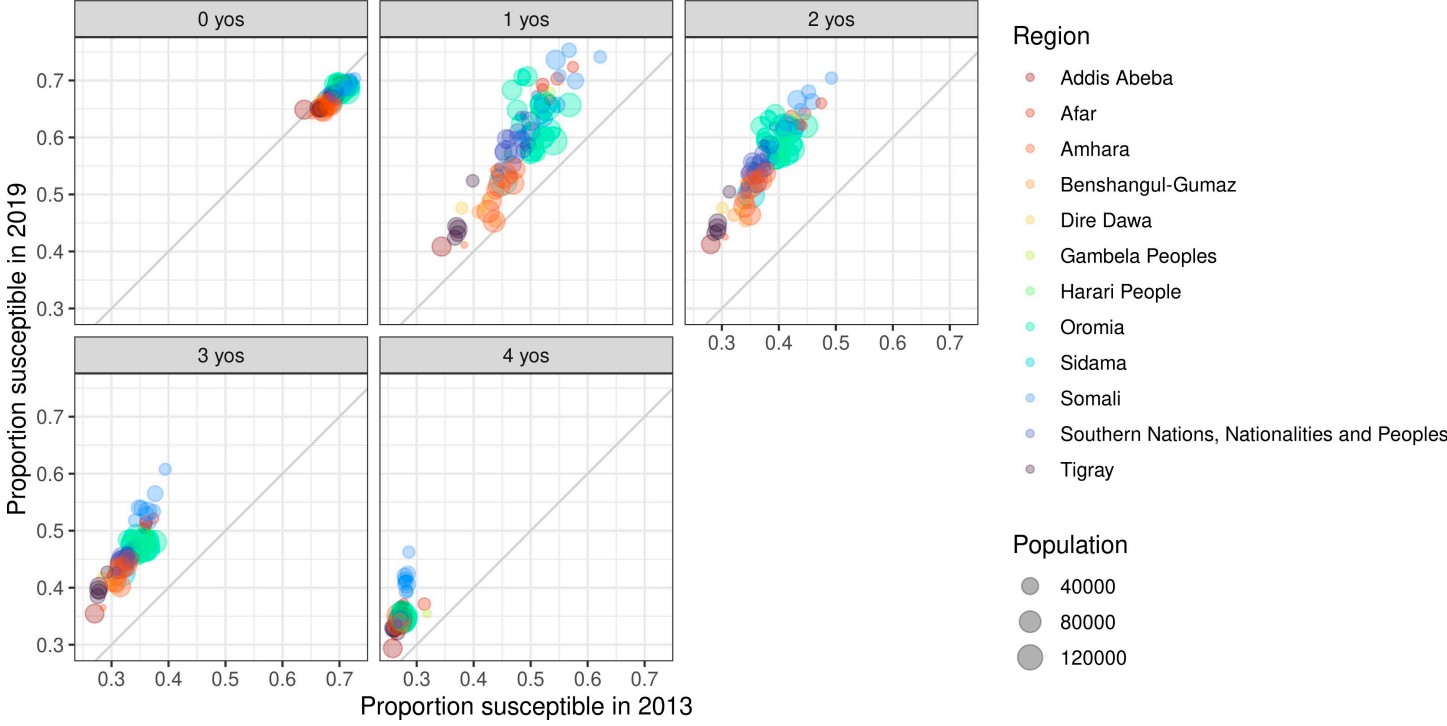

**Fig 11. Susceptibility by age (among 0-to-4-year-olds) in 2013 compared to 2019.** Each point represents a zone, is coloured by region and is sized by population.

vaccination, natural infection, and maternal antibodies. Additionally, vaccine coverage data do not present the full picture of susceptibility and also require assumptions about doses administered through SIAs (which also may be incorrectly reported or estimated), and, finally, case notifications are often under-reported. We explored the utility of using case notifications in subnational transmission models and validity of modelling approaches to estimate susceptibility and case ascertainment. We faced challenges related to data quality

and statistical and computational complexity. While we used the example of Ethiopia as a case study, we expect the methodological considerations, challenges faced both in data interpretation and model fitting, and overall discussion on these topics to be widely generalizable to other settings with similar surveillance systems and data availability. With appropriate and similar input data as used for Ethiopia in this work and in-country expertise to guide parameter assumptions (e.g., a range of plausible reporting rates, vaccine effectiveness values, probability of staying in home subnational unit in a given week), our model could easily be applied to other locations.

Overall, our experience suggests that several modelling considerations are critical when fitting models to estimate subnational susceptibility. First, underlying trends in reported case data may be biased in multiple ways (e.g., sporadic reporting, incomplete ascertainment). Without accounting for these biases, estimates of susceptibility are likely to be themselves biased. Additionally, when highly granular case notification data are available, high-dimensional models are challenging to fit using traditional Bayesian methods. If Bayesian methods fail to converge, modellers may need to explore alternative deterministic optimization fitting approaches. Finally, assuming that vaccine effectiveness is equivalent to ideal vaccine efficacy can skew epidemiologic conclusions and produce results that do not account for underlying mechanisms to lower effectiveness that should be interpreted with caution.

We faced substantial challenges fitting our model via statistical algorithms. Many data observations from reported case notifications suggested a large deal of artificial certainty that did not account for unmeasured uncertainties, which created a steep likelihood surface. In response, we attempted to various ways of accounting for this concern, including noise removal (i.e., smoothing), exploring various distributional assumptions (e.g., negative binomial), allowing for flexible reporting rates, and using various adaptive fitting mechanisms; these were all only partially successful.

If our dataset were smaller, we ran our model for fewer years, locations or age groups, or explored substantial adaptations to our sampling algorithm [40], we may have been able to use Bayesian methods for model fitting. However, these modifications will also remove the ability to estimate susceptibility within small areas and age groups. Hence, we instead moved forward with a deterministic optimization approach via block coordinate descent. This approach was more computationally efficient and yielded consistent convergence, as it was faster to optimize just one parameter at a time. This however limited our exploration of the joint parameter space.

Understanding the mechanism for case reporting is critical for understanding how best these case data could be incorporated into fitting dynamic transmission models. Experience from Ethiopia suggests that these mechanisms are very varied, but suffer from location-, and time-dependent biases, which reduces their information content. Therefore, we had to make various assumptions. For example, due to frequent one-week gaps in case reporting as well as otherwise sporadic reporting, we assumed that cases were inconsistently reported temporally so chose to smooth cases before including them in model fits. We estimated case ascertainment rates of less than 3% across many regions. Robust surveillance systems are the backbone to understanding measles dynamics and as countries approach elimination are even more essential to understanding remaining gaps in vaccination programs.

Our sensitivity analysis on vaccine effectiveness suggested that lower values than typically assumed (i.e., 47%) yielded better model fits (via AIC score), as well as $R_0$ values that were lower than often-cited ranges for measles [38,39] but suggested as plausible by other sources [37]. While these lower vaccine effectiveness metrics could reflect true low effectiveness stemming from disrupted cold chains or lack of seroconversion due to individual biological factors, such as malnutrition or compromised immune function due to human immunodeficiency

virus (HIV) infection, this metric could also be reflecting other uncertainties in our model. One alternative option is that coverage from both RI or SIAs might not be as high as originally estimated or that there are limitations in the current available data, which are aggregated to zones, to capture true coverage heterogeneity. If this is occurring, the lower vaccine effectiveness measure might be adjusting coverage to reflect these potential inaccuracies.

In addition to vaccine effectiveness, the concurrent estimation of under-reporting and force of infection is similarly challenging. For example, policymakers could exhibit caution regarding a potential measles outbreak. Consequently, they may choose to plan a large-scale SIA, which could significantly increase coverage, and simultaneously could intensify surveillance efforts. These developments, if implemented together, could subsequently complicate the correlation between coverage and reported cases. Additionally, there may be either geographic or social clustering of measles vaccine coverage so that there are large pockets of highly connected unvaccinated persons even within zones. Any one or multiple of these could be occurring in these dynamics, and without a way to stay with confidence which is occurring, this vaccine effectiveness metric, in combination with other parameter values, should be interpreted with caution.

Our models suggested higher susceptibility broadly in 2019 relative to earlier years, likely stemming from overall low MCV1 coverage and a lack of recent vaccination campaigns with few recent widescale outbreaks. Given the unfolding situation in Ethiopia in years since 2019, these results suggest residual susceptibility gaps that leave many vulnerable children susceptible to preventable disease and death. Recently, in 2022, increased measles cases have been reported across 44 woredas (i.e., third-administrative units) among eight regions in Ethiopia [41], particularly in the Somali region and the Gamo zone of the former Southern Nations, Nationalities and Peoples (SNNP) region [42]. Gamo zone is one of the most populated yet rural zones within the country and located in the southwest. During our last year of estimation (i.e., 2019), our models predicted higher susceptibility among the Somali region, but did not predict higher susceptibility in the Gamo zone relative to other zones in the region. This could, in part, be due to additional factors related to vaccination program distribution attributable to the SARS-CoV-2 pandemic [43] or other relative changes in person mobility or contact related to the pandemic. Future work will be needed to incorporate these effects as well as other factors that could decrease coverage or case reporting, increase susceptibility, and heighten disease risk in the years since the end of the study period (e.g., conflict, displacement). Then, future investigations could assess the ability of our model to predict these recent outbreaks. Models, including those presented in this work, would require validation prior to their use to guide operational decisions. However, these validations remain challenging in light of evolving epidemiologic and socio-political circumstances and in the absence of gold-standard data to validate models against, including a lack of available recent seroprevalence data. Identifying opportunities for enhanced model validation will be critical, though, for the uptake of these frameworks for decision-making.

There are several limitations to our analysis. First, we did not have access to stratified cases beyond five-year age bins nor were we able to distinguish between locations that reported zero cases as true zeros or a lack of reporting. We only considered suspected measles cases and did not consider lab-confirmed measles cases or adjust for test positivity rates. Therefore, the cases we considered in this model may have been misclassified as measles instead of, for example, rubella. As rapid diagnostic tests for measles and rubella are still under development and programmatic evaluation, many locations, including those that experience simultaneous measles and rubella burden, do not currently have access to adequate confirmatory testing for suspected measles cases. Therefore, in order to mitigate rubella misclassification, approaches that only consider measles IgM positive cases may miss a substantial proportion of true measles

cases. More exploration is necessary to examine whether there is any additional information that can be learned about reporting rates and possibly increased usability of case notifications once rapid diagnostics become available for widescale use. We did not have stratified contact patterns among age groups smaller than 5-year bins, which limits our understanding particularly among contact patterns in 0-to-4-year-olds. This age group is likely to have substantial heterogeneity across the age group (i.e., infants versus 4-year-olds) as well as the age group in which we would expect the most measles transmission to occur. We did not have access to mobility data so assumed the proportion of persons staying home ( $\psi$ ), although this parameter ideally would be generated from inference. There are two serosurveys available in Ethiopia [4] from 1994 and 1999, however we did not use these in our model fitting as their relative contribution to the overall magnitude of our likelihood was likely to be negligible. Additionally, in the absence of recent seroprevalence data or alternative data sources, it was challenging to validate the results our modelled results in Ethiopia to recent real-world trends.

We used a gravity model to estimate person mobility via motorized transport; however, this might not be an accurate representation of connectivity within Ethiopia. Therefore, model adaptations with alternative spatial dynamics should be explored. These could include use of travel time metrics for persons traveling on foot, collecting data from road maps, or other model structures such as radiation models. We were unable to effectively estimate vaccine effectiveness outside a sensitivity analysis as the collinearity of all three sets of parameters (i.e., also maximum and minimum transmission parameters, and reporting parameters) was challenging for model fitting; additional analyses could further explore a more precise range for the vaccine effectiveness measure. Also, although we assumed a single national vaccine effectiveness parameter, it is possible that vaccine effectiveness is heterogeneous across space, related to subnational variation in cold-chain problems and malnutrition, among other factors. However, given the challenges in fitting the model described, additional data would likely be needed – or other parameters would need to be assumed – to allow for a model that captures this additional level of detail to be identifiable. The uncertainty in our estimates was obtained from our bootstrapped samples, which only reflect uncertainty from the data alone.

Our model estimates at the zone-level (i.e., second-administrative unit), which may not be able to accurately capture sub-zone level local transmission that may persist in hyper-localized or non-geographic social networks. Complementary field-based approaches, coupled with model types suitable for these data such as network or agent-based models, to assess local heterogeneity in immunity may be appropriate in settings with high coverage or communities experiencing challenges in immunization access or confidence. Finally, our exploration of cases was of those reported in Ethiopia through the end of 2019, which does not consider changing epidemiology, reporting patterns, or demographic, contact or mobility changes associated with the COVID-19 pandemic [43], ongoing conflict and insecurity [44], famine [45] and drought. Additional future work is needed to further explore plausible methods to elucidate subnational measles susceptibility given these evolving transmission dynamics and available data landscape.

There are several practical limitations to our analysis as well, that could prevent this work from being expanded more broadly and attempted in local contexts. We note several computational challenges that we were able to explore as we had access to substantial computational resources and a distributed computing environment. Our proposed model fitting solution yielded more reasonable computational times for fitting models in Ethiopia but still took approximately 5 hours to finish despite being run in a distributed computing environment; this approach may not be feasible in a local computing environment. Additionally, this model in Ethiopia only had 79 zones; many other countries, such as Nigeria or India, have 10 times

or more second-administrative units. While this model is generalizable to other settings, practical considerations around resources to scale this model to other locations would need to be considered. Additionally, we acknowledge the current scope of resources (e.g., time, expertise) in many high burden settings and the subsequent challenges that adapting these methods in these local contexts. Given these limitations broadly along with ongoing operational practices (i.e., making prioritization decisions based on triangulating estimates of vaccination coverage), modelled results of this kind currently are best suited to be used alongside existing processes and data streams as an additional resource for decision-makers.

Understanding subnational susceptibility is essential to contribute towards operational decisions for planning targeted interventions. Our analysis highlights the technical challenges of using case notifications and highlights data limitations that may need to be addressed to improve the ability of models to identify subnational susceptibility gaps.

## Supporting information

**S1 Text.   Contains additional supplemental methods, tables and figures**.
(DOCX)

## Acknowledgements

We thank Bradley Bell, Stefan Flasche, Han Fu, Bobby Reiner, and Naomi Waterlow for discussing model fitting methods.

## Author contributions

**Conceptualization:** Alyssa N. Sbarra, Mark Jit, Jonathan F. Mosser.

**Data curation:** Jason Q. Nguyen, Rebecca E. Ramshaw, Sam Rolfe.

**Formal analysis:** Alyssa N. Sbarra.

**Investigation:** Alyssa N. Sbarra, Emily Haeuser, Samuel Kidane, Andargie Abate, Ayele M. Abebe, Muktar Ahmed, Tsegaye Alemayehu, Erkihun Amsalu, Akeza A. Asgedom, Nebiyou Bayleyegn, Mulat Dagnew, Biniyam Demisse, Werku Etafa, Getahun Fetensa, Teferi G. Gebremeskel, Habtamu Geremew, Abraham T. Gizaw, Gamechu A. Hunde, Hadush N. Meles, Sibhat Migbar, Eshetu Nigussie, Biniyam Sahiledengle, Noga Shalev, Yonatan Solomon, Latera Tesfaye, Gesila E. Yesera, Mark Jit, Jonathan F. Mosser.

**Methodology:** Alyssa N. Sbarra, Emily Haeuser, Aleksandr Y. Aravkin, Mark Jit, Jonathan F. Mosser.

**Software:** Alyssa N. Sbarra.

**Supervision:** Mark Jit, Jonathan F. Mosser.

**Validation:** Alyssa N. Sbarra.

**Visualization:** Alyssa N. Sbarra.

**Writing – original draft:** Alyssa N. Sbarra.

**Writing – review & editing:** Emily Haeuser, Samuel Kidane, Andargie Abate, Ayele M. Abebe, Muktar Ahmed, Tsegaye Alemayehu, Erkihun Amsalu, Aleksandr Y. Aravkin, Akeza A. Asgedom, Nebiyou Bayleyegn, Mulat Dagnew, Biniyam Demisse, Werku Etafa, Getahun Fetensa, Teferi G. Gebremeskel, Habtamu Geremew, Abraham T. Gizaw, Gamechu A. Hunde, Hadush N. Meles, Sibhat Migbar, Jason Q Nguyen, Eshetu Nigussie, Rebecca E Ramshaw, Sam Rolfe, Biniyam Sahiledengle, Noga Shalev, Yonatan Solomon, Latera Tesfaye, Gesila E. Yesera, Mark Jit, Jonathan F. Mosser.

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
