## [Decision Letter · Decision Letter 0]

2 May 2024

Dear Ms. Sbarra,

Thank you very much for submitting your manuscript "Exploring the utility of subnational case notifications in fitting dynamic measles models in Ethiopia" for consideration at PLOS Computational Biology.

As with all papers reviewed by the journal, your manuscript was reviewed by members of the editorial board and by several independent reviewers. In light of the reviews (below this email), we can not accept your MS for publication in its actual form. However, I would like to invite the resubmission of a significantly-revised version that takes into account the reviewers' comments.

We cannot make any decision about publication until we have seen the revised manuscript and your response to the reviewers' comments. Your revised manuscript will likely be sent to reviewers for further evaluation, although if we appreciate that the revision is not substantial enough given the feedback received, we could also make a decision without further review.

Sincerely,

Yamir Moreno

Academic Editor

PLOS Computational Biology

Virginia Pitzer

Section Editor

PLOS Computational Biology

Reviewer's Responses to Questions

**Comments to the Authors:**

Reviewer #1: Thank you for the opportunity to review this manuscript.

The authors wish to understand the risk of potential index cases at subnational level to determine where to deploy vaccination. The specific vaccination strategy to be considered in response to these cases is not mentioned but given the data sources used is likely to be targeted follow up mass vaccination rather than pulse vaccination. The authors focus on computational considerations that are to be made in fitting and estimating from their model. The main outcome of interest is the number of susceptible persons at various administrative levels.

Measles is an infectious disease of public health concern as a) it requires a very high vaccination coverage to achieve herd immunity (normally in the 95 per cent range), b) it has a high basic reproduction number (usual estimates are around 13), and c) it can cause immune amnesia putting survivors more at risk of other infections afterwards. However, measles (exposure >and< survival or vaccination) should confer lifelong immunity (which allows for simplifying assumptions in a modelling approach, e.g. there need be no flows from removed back to susceptible) and it is only assumed to have a human reservoir making it a candidate for elimination/eradication. There is a dependency in vaccination that needs be captured, namely that second dose is contingent on having received first dose.

The WHO immunisation data portal suggests the routine measles immunisation schedule for an Ethiopian infant is first dose at nine months and second dose at fifteen months. I unfortunately do not know much else about the situation in Ethiopia, but I believe it is bordered by countries with a certain amount of instability whereby regions near the borders could be of particular concern.

With some improvements, I believe the manuscript can become fit for publication. I would be willing to review a revised version of the manuscript. I do not consent to this review being unblinded.

Wording I found particularly pleasing to read:

The first two sentences of the abstract set the scene and provide the motivation for the work in a nice manner. The first sentence of the paragraph on line 136 is nice as well as the last sentence in the paragraph on line 513.

I would suggest updating the notation to ensure avoiding the same symbols for different things, examples:

* g_zy = P_z P_y / f_{zy} (avoids mixup of i)

* m_zy = \begin{cases} (avoids mixup with M compartment)

\frac{g_{zy}(1-\xi)}{\sum_y g_{zy}} & z\neq y \\ (avoids mixup with theta used in supp. inf. and D used for phase)

\xi & z = y \\

\end{cases}

* \xi \in [0, 1], \xi \equiv 0.99

* \beta_w = A \sin\left(\frac{2\pi w}{52} + 2\right) + D, w = \text{week} (avoids mixup of i and a)

* FOI_{adw} = \beta_w \sum_c \sum_y (W_{acwy} m_{zy} I^\alpha_{w-1,cz}, \alpha \in [0, 1], \alpha \equiv 0.99

It may be beneficial to the authors as well as the reader to provide an overview table with the variables used and their meaning as can be found in engineering disciplines. Reading the manuscript backwards may help the authors realise where terminology varies (e.g. time step and week) and allow them to harmonise it.

Additional comments:

Line 108: Providing the numbers for the other years in the study period (i.e. national MCV1 coverage for 2013 to 2018, both years included) will assist the reader in understanding the situation and will support and quantify the statement on line 111: “increased national MCV1 coverage”

Line 116: Providing a population estimate will help the viewer understand the magnitude of this number

Line 123: It is unclear why the United States is being appealed to here and I suggest this either be explained or removed. “Different strategies” is also vague and could be improved by considering specific strategies, e.g. pulse vaccination is often used for measles, but ring vaccination could also be used

Line 133: If serosurveys are not being used in the work, I would remove this sentence

Line 153: This information could perhaps be placed in a table with additional information such as study period and location considered

Line 167: I would say “compartmental modelling” rather than “traditional modelling”

Line 180: Perhaps a table will help the reader understand the administrative divisions of Ethiopia, something like:

+--------------------------------------+--------+

| Level | Number |

+--------------------------------------+--------+

| Country | 1 |

+--------------------------------------+--------+

| Regional states and chartered cities | ~11 |

+--------------------------------------+--------+

| Zones | ~79 |

+--------------------------------------+--------+

| Districts/woredas | ~700+ |

+--------------------------------------+--------+

Line 181: Abbreviation never used again after introduction so not needed

Line 189: What does the spatial autocorrelation of cases look like at zone level?

Line 192: Abbreviation SIA is already introduced in the introduction section

Line 195: Perhaps the authors should note that “epiweek” is an ISO week shifted one day backwards such that the week starts on Sunday rather than Monday as this might not be common knowledge depending on the cultural background of the reader

Line 197: The “additional details” also references previous work so please at least provide a short description. Not all readers may be able to access the Nature reference

Line 200: Why is the model relegated to supp. inf. rather than included in the main manuscript? Please add the model equations also as it is unclear why no model equations are given and flows in the diagram are not given parameters

* Additionally, does it make sense to distinguish R_unvax, R_MCV1, and R_MCV2 in your compartmental model? Given that these compartments cannot become infectious perhaps a simplifying assumption would be to have just a single compartment for removed and examine the size of the flows into this compartment. The public health implication should be—but is likely limited by resources—that if it is possible to first determine antibodies, only those with no prior measles infection should be targeted for vaccination. I would suggest including the MSIR model equations in the main manuscript such that the reader knows the assumptions being made in the work

Lines 202 and 231: This seems inconsistent with study period and age groupings provided previously

Line 209: Are other countries than Ethiopia being considered?

Line 209: A comparison with on contacts in South West Shewa Zone of the Oromia Region by Trentini et al. (2021) https://doi.org/10.1186/s12916-021-01967-w would be nice

Line 215: Suggest reference to Xia et al. (2004) https://doi.org/10.1086/422341

Line 229: Would be better to include the MSIR compartmental model equations in the manuscript

Line 233: Unclear why this is relegated to supp. inf.

Line 238: “some” is vague, please quantify

Line 239: This is also vague, please qualify what exactly you tested and what “little to no difference” means

Line 243: Should note that beta is the flow from the susceptible to infected compartment (assuming common notation used)

Line 256: Provide context, is late March significant in Ethiopian culture or is this common to the measles virus?

Line 272: Abbreviations already introduced not used

Line 275: Vague, please note which algorithms specifically

Line 279: Why was 5 chosen?

Line 281: Inconsistent index for week, please see earlier suggestion on updating notation

Line 283: Please provide version for package used for reproducibility

Line 287: No evidence is provided to support this statement, please add

Line 292: Please examine this, what are the results of modelling fortnightly rather than weekly and how does it differ from your original results?

Line 302: Note which goodness-of-fit metric is being used

Line 308: LOESS should be capitalised as it is an abbreviation

Line 315: Suggest adding the reference underlying this assumption here also (reference 30 cf. line 324)

Line 318: Provide which “more recent years” are being referenced

Line 320: Please provide the notation for this parameter so the reader can follow it

Line 332: Please compare with other methods such as hidden Markov chains and nowcasting

Line 335 and 340: Should this not be d rather than R (or d\in R) and is not this d for zone as previously implied in equation for force of infection?

Line 357: Please provide a table with performance metrics

Line 365: Please provide details of computational environment as this can have an influence

Line 367: Please provide stopping criteria and thresholds for algorithms used for reproducibility

Line 376: Please provide package version and settings used for reproducibility

Line 377: Please quantify the fraction for comparison with previous estimate of 20 seconds

Line 389: I don’t think this is sensitivity analysis, which consists of checking robustness of results to assumptions made, but rather something like a profile likelihood two-stage estimation approach

Line 401: This is inconsistent formatting with previous packages mentioned. Please provide a package version number

Line 402: Provide details of “initial starting state” for reproducibility

Line 405: Provide convergence/optimisation settings

Line 409: MLE (as abbreviation is already introduced)

Line 418: Is 100 samples enough? The data used is large and this number is small

Line 433: Figure 1 does not show zones. For comparison, standardised rates to account for population size may make sense for Figure 1

Line 434: Please note that this is 364 weeks in total since that is the time unit being used (in addition to not instead of)

Line 435: A graphical representation would help the reader

Line 442: Please provide intuition on which source is more likely to be correct. Corrections of time series for cases arising from different systems may have been considered during COVID-19 and may be something the authors could investigate

Line 449: Please explain why endemic transmission would not be expected in Ethiopia when you have previously explained that the second dose is only introduced a few years ago (at time of writing)

Line 467: This came out of nowhere, please introduce the socio-demographic index in the methods section

Line 471: I would suggest discussion of the distribution of remaining susceptible cases be the first part of the results section as that was the exposition given in the introduction

Line 478: Please quantify the “negligible differences”

Line 483: Please describe when, where, and which format the mass campaign events have for the unaware reader

Line 488: Should context be given that Somali region borders Somalia which is politically unstable?

Line 549: RI (abbreviation already introduced)

Line 570: Should context be given for the Gamo zone? Should this be zone? It is inconsistent with previous “region” in the sentence

Line 577: What about displacement? This is a known driver as well as a known humanitarian health issue in Ethiopia

Line 584: But does demographic data exist on which to construct a synthetic contact matrix for these ages? Also, were they not included in the 1980 to 2019 window?

Line 597: The authors could also consider use of road maps

Line 617: Rouge full stop

Line 619: Should presumably be https://github.com/alyssasbarra/ethiopia_case_fitting

Line 620: This is not possible to review as it is not available

Line 749: The authors should note in the caption that differing axes are being used

Line 776: The colours in the figure are difficult to distinguish in black and white, suggest use of different line types as well as annotating (could be geom_rect) where the study period is (2013 to 2019)

Line 781: Discrepancy between reported suspected and reported in caption and axis label

Line 793: Same comment as previously. The authors should note in the caption that differing axes are being used. The authors could consider adding vertical lines for the thresholds considered in the work (88 per cent and 47 per cent and so on)

Line 811: Suggest reversing the colour scale as most might expect a darker colour to be a greater value. The colours in the figure are difficult to distinguish in black and white and the borders are difficult to see for the darker colour

Line 827: Please comment on the greater proportion susceptible found in 2019 and the public health implications of this

Line 831: Please note points are sized according to population count

Regarding the supporting information:

* As a general comment, it is unclear why the model and results have been relegated to the supporting information. As far as I am aware, PLOS Computational Biology does not have restrictions on number of figures in manuscripts, so it would make more sense to include them where they are discussed in the main text.

* LASSO should be capitalised as it is an abbreviation

* Note package version for TMB package

* Suggest adding lines and labels to distinguish country (C_{t,g}, \logit(p_{t,g}), region (C_{t,c} etc.), and zone (C_Pt,a_1}). It seems there is a copy-paste error in the third binomial

* Which type of splines are being used?

* “Methods have been described elsewhere” is not sufficient

* Should the campaign efficiency p not contain the risk for the opposite outcome in the denominator?

* Please describe how demographic change is incorporated

* All figures are difficult to distinguish in black and white

* Please index figures by dates from 2013 to 2019 rather than numbers or provide a secondary x-axis with the date

* Table A1: Page number are not included so suggest adding them or changing the heading to say “section”

* Should the tables not say Ababa rather than Abeba?

* Table A2: Please emphasise where in the table the inconsistencies are to aid the reader (e.g. with bolded font)

* Table A3: Please write log-likelihood for consistency wit following table

* Please comment on why there is absolutely no uncertainty in estimates in Table A4 (from row 4 onwards it is estimates of 1 with an interval of 1 to 1, which seems unreasonable). Is this indicative of a convergence problem?

* Figure 1A: What is appendix B section 1? Is there a second supporting information? If so please combine them rather than sending the reader back and forth between different documents

* Figure A2: Please use dotted and dashed to avoid confusion. Please annotate the flows (currently most would be assumed to be the same without additional information given, e.g. subscripts) and provide the model equations

* Figure A3: it might be easier to distinguish model fit as a curve and observed cases as points. Why is this region specifically highlighted?

* Figures A4 and A5: Please improve the labels

* Figure A5: Please note in the caption different axes are being used

* Figure A6: It is unclear what the added value of this figure is, the linearity could be mentioned in a sentence in the text

* Figures A8, A9, A10, A11, A12, and A13: Please enlarge, these are too small to read or scrape. Consider using all the space on the page as well as allowing to go on separate pages

Regarding the references:

* Formatting of references is inconsistent with respect to use of ISO4 abbreviations and capitalisation of PLOS. I suggest authors harmonise their references

* No DOIs are given in the bibliography

* Is Reference 19 different than Reference 4?

* Unclear what “14th” in reference 30 means

* For reference 39 why not link to arXiv itself rather than a Harvard page?

Regarding PLOS Computational Biology specifically:

* The manuscript does not currently adhere to the PLOS’s editorial and publishing policies (https://journals.plos.org/ploscompbiol/s/editorial-and-publishing-policies). In particular, the data availability is not fulfilled which is also the case for point 8 in their Table A1. As the data is aggregated by age group there should not be identifiability issues whereby the authors should release it such that other researchers can recreate/reproduce their work.

* Regarding authorship, the authors should use the CRediT Taxonomogy as adopted by PLOS (https://journals.plos.org/ploscompbiol/s/authorship) which will provide transparency regarding contribution and suggests that R. Ramshaw, S. Rolfe, and J. Nguyen be upgraded from acknowledgement to author (based on the description of their contributions given in lines 624 to 626). Additionally, the authors could consider a different ordering of authors that allows the in-country researchers to take the prestigious first and last position cf. Nassiri-Ansari et al. (2024) https://doi.org/10.37941/RR/2024/1

Regarding the code repository:

* Contains no file suggesting which order to run code in, some are numbered and some not; adding a README file would improve usability greatly. Unit tests, documentation, and linting would also improve the code. As R is a functional programming language use of functions in conjunction with apply rather than loops should speed up the code for the R scripts.

* Note that certain auxiliary functions seem to be out of limits in the code repository, scripts such as “setup.R”, “misc_vaccine_functions.R” and “get_population.R” are not provided meaning other users may not be able to use the code as provided. I would suggest the use of a container to ensure correct versions of packages are used (as R has known backwards compatibility issues) or at least noting them in the manuscript.

* Assuming use of ggplot, if the authors wish to ensure their plots all have the same look (e.g. figures A4 and A5 in the supp. inf. look different), they could consider use of theme_set()

Reviewer #2: Recent years have seen a mass resurgence of measles throughout the world with low and middle income countries (LMICs) being hit particularly hard in terms of morbidity and mortality. Measles susceptibility reconstruction, namely the attempt to estimate latent (typically age and space structured) susceptibility of a population to measles infection, is therefore a very timely topic of substantial relevance to both modelling and operational communities.

Assessment of susceptibility, however, is nontrivial and there is an interest in exploring potential new approaches to improve reconstruction. The authors contribute an additional modelling approach that complements other work already being done in the broader modelling community. Their approach attempts to leverage a substantial amount of data (notably suspected case data, vaccine coverage data, and population data) in hopes of explicitly quantifying susceptibility.

Generally, the paper is methodologically interesting and raises good points about some of the challenges of highly parameterized models in the face of extreme data frailty. Unfortunately, the overall goal of the paper is unclear. It is difficult to tell if the paper aims to be a discussion of methods and challenges (which would be interesting and valuable) or if it aims to present the results and (presumably operational) recommendations pursuant to this particular implementation. If the goal is the latter, the authors may need to dramatically expand the evaluation and operational interpretation of the model. In either case, the paper would benefit from having a clearer stance on what it is trying to accomplish.

If the authors opt for a methods focus, I think the paper (with a bit more information and discussion placing their model into context with other similar methods) could be a valuable contribution. Particularly as many groups may be interested in pursuing direct quantification methods for susceptibility reconstruction (as opposed to prioritisation oriented methods) it would be useful to have a publication highlighting the challenges such an approach may face.

---

Particularly positive aspects:

1. The methods explanation, while dense, is very detailed and I greatly appreciated the transparent discussion of challenges the team faced in fitting as well as the very complete discussion of what strategies were tested (as opposed to only reporting the final method used).

2. The conversation of computational cost is an important one (particularly for resource constrained settings) an it is interesting to see methods that explore ways to leverage bootstrapping to create deterministic (and therefore computationally cheap) methods that retain a notion of uncertainty; while this uncertainty is somewhat "constructed", this is an issue that the authors clearly acknowledge and I do not see an apparent better solution than what they have proposed.

3. Generally the discussion of limitations surrounding mobility modelling, uncertainty, and certain data frailty issues is thoughtful and well appreciated.

4. Great to see a large number of collaborators/authors from the study country (here Ethiopia) on the paper !

5. Glad to see the commitment to open source code.

A more neutral point:

1. Case ascertainment of 3%. While we know that measles is massively underreported, 3% ascertainment is very concerning (though this is not to say the estimate is necessarily unrealistic). It would be fascinating to see other work looking into periods of active followup (though I am not aware of anything published for Ethiopia) or an analysis of potential reporting shifts seen during a period of active case finding (for example during a riposte) if there are any in the historic data.

Some things to consider:

1. Data sources. It is not clear where a lot of this data comes from and it is therefore difficult to assess its associated reliability. For example, MCV1 coverage estimates can vary dramatically depending on where you get them from. It would be interesting to know if the authors are using administrative/independent estimates and if they compared multiple sources of coverage data to investigate quality. Similarly, it would have been nice to see a check of whether the coverage negatively correlated with epidemic risk (as would be expected). While I very much appreciate that the authors take time to discuss data quality (particularly for case data), a bit more nuance would be helpful. For example, the authors note that they don't see the expected negative correlation between suspected cases and vaccine coverage and conclude that this likely speaks to the poor quality of case data. It is equally possible, however this result has more to do with issues of vaccine effectiveness and/or issues in the coverage data (which can sometimes rely on highly frail and inaccurate population estimates).

2. Operational purpose. The authors suggest that this model will somehow be used to inform SIAs, though how that would be done and why explicit quantification of susceptibility (as opposed to prioritisation) is required is unclear. Explicit quantification is challenging and providing an explanation of why this may be worth the additional modelling effort over a ranking/risk oriented model would be useful. If, following the challenges faced, the authors conclude that perhaps a prioritisation approach would have been equally good or better, this should also be discussed.

3. On SIAs. The authors do not seem to have a perfectly stable definition of who SIAs should target, line 119 suggests SIAs should target areas with moderate coverage but 136 indicates that targeted interventions should look for vulnerable (unprotected) persons. It is also worth noting that claim of line 119 is highly debatable and ignores the fact that SIAs and other targeted interventions in LMICs will often face resource constraints limiting the number of people vaccinated. In these cases, one might elect to target certain priority groups, often those facing highest risk of mortality (children <5), even when a broader age target may do a better job at reducing overall transmission.

4. Complexity. The proposed model is quite complex, requiring several assumptions and facing potential problems of parameter identifiability pursuant issues of multicollinearity. While similar challenges likely face other modelling approaches, some of the assumptions may warrant some additional justification (for example, theta is set at 0.99 with no explanation) and or at minimum some discussion. For example, the authors choose to bring vaccine effectiveness into their model, rightfully noting that it is subject to being undercut by issues of cold chain breakdown and/or high prevalence of malnutrition. Cold chain problems and malnutrition, however, are often characterized by substantial spatial heterogeneity, which should presumably then be reflected in the model but does not seem to be. It would also be interesting to see more discussion of whether the authors believe all assumptions were ultimately necessary for adequate model performance. It may also be interesting to consider the degree to which the model complexity (in addition to the artificial certainty of the data, as noted by the authors) may have contributed to the steepness of the likelihood surface.

5. Issues of generalizability. It is unclear if this model is expected to be generalizable to other settings and what adjustments would be needed to adapt it (for example in the assumptions and fixed parameters like vaccine effectiveness). If the goal is operational recommendation/impact, a discussion of generalizability is warranted.

6. Model validation. While it is nice to see the use of the AIC for model comparison, it would have been valuable to see some work done on model validation, particularly if the goal is operational impact. The authors do this very slightly in their discussion of increasing trends in the Somali region and Gamo zone, but more work is warranted to give a proper explanation of whether the susceptibility estimates of the model make sense in light of real world trends. While the authors correctly raise concerns about the quality of suspected case data, which may make it difficult to interpret correlations (or not) between the estimates and case numbers, they could still for example, look at comparing with something like general epidemic risk. Epidemic risk will of course rely on case data, but should be meaningfully more robust.

7. Broader operational considerations. If the goal of models like this is to inform operational decision making, some consideration should be given to the anticipated challenges/feasibility of real world implementation. For example, is it realistic to implement such a model in an LMIC long term? Is the model worth implementing over a less complex model that can only provide a prioritisation output? Or, even further, since model susceptibility estimates correlated with vaccine coverage estimates, it would be good to discuss why operational decision makers should use this model over a strategy that simply chooses to target age groups / areas with the lowest estimated coverage (which is what many countries currently do).

8. Rubella. No discussion is given to how case misclassification (particularly with rubella) may have impacted the outcome of the model. It would be interesting to see a discussion of whether the model is equally generalizable to settings with high/low rubella burden, particularly in areas using the monoantigen measles vaccine.

One other small issue, the authors state they have used R version 5.4.0, which is presumably a typo for 4.4.0 (the latest current version as of March 2024).

Reviewer #3: Attachment

**Have the authors made all data and (if applicable) computational code underlying the findings in their manuscript fully available?**

Reviewer #1: **No: ** Data is not available and code cannot be run without additional scripts

Reviewer #2: **No: ** The authors have made their code available but not the data. This is expected as they are using health related data.

Reviewer #3: None

PLOS authors have the option to publish the peer review history of their article (what does this mean? ). If published, this will include your full peer review and any attached files.

**Do you want your identity to be public for this peer review?** For information about this choice, including consent withdrawal, please see our Privacy Policy .

Reviewer #1: No

Reviewer #2: **Yes: ** Catherine Eisenhauer

Reviewer #3: No
---

## [Decision Letter · Decision Letter 1]

5 Sep 2024

Dear Ms. Sbarra,

Thank you very much for submitting your revised manuscript "Exploring the utility of subnational case notifications in fitting dynamic measles models in Ethiopia" for consideration at PLOS Computational Biology.

Your manuscript was reviewed again by the editors and 2 of the previous reviewers. In light of the reviews (below this email), we cannot accept your contribution for publication in PCB. However, we could consider a revised version that takes into account the reviewers' comments through a major revision. Please, note that your MS has been reviewed two times and we would like to keep the number of review rounds to the minimum possible, thus, should you decide to resubmit the MS, this would be the final round of revision (except for very minor changes remaining), after which, we will make a decision.

We cannot make any decision about publication until we have seen the revised manuscript and your response to the reviewers' comments. Your revised manuscript is also likely to be sent to reviewers for further evaluation.

Sincerely,

Yamir Moreno

Academic Editor

PLOS Computational Biology

Virginia Pitzer

Section Editor

PLOS Computational Biology

Reviewer's Responses to Questions

**Comments to the Authors:**

Reviewer #1: It is unclear to me why the authors would both suggest that the code repository is incorrect yet provide a README for a repository which is not the one used. I am unable to review the code but am otherwise satisfied with the responses given. Apologies for the short review but I am slightly pressed for time currently.

Reviewer #2: This is a second review of the manuscript Exploring the utility of subnational case notifications in fitting dynamic measles models in Ethiopia. As stated in my initial review, this paper tackles the relevant and timely question of whether modeling efforts can be used to estimate subnational measles susceptibility in moderate burden areas, here Ethiopia.

In my initial review I commented that the paper lacked clear direction, seeming to be in between a methods paper on the challenges of building this type of model in data frail environments and a paper more focused on the direct results of the specific implementation for the Ethiopian context. I recommended that the Authors shift towards the first of these options, focusing on both the difficulties faced and the solutions that they were able to find (which were well thought out and clever). In this new version of the paper, the Authors have generally taken this advice, characterizing the work as a case study.

While this shift in focus is appreciated, the paper does not seem to have fully transitioned to a methodological focus. The results section remains oriented towards the specific outputs of the given implementation rather than a discussion of the methodological challenges themselves, which could have taken center stage were the paper to be more centralized around how to develop models in similar contexts.

Points appreciated in the revision:

- In the methods section, the Authors have done a good job providing improved explanation of the model and its various aspects; for example explanation of the computational framework required, the choice of certain statistical parameters, and the choice of constant values.

- In particular, expanded explanation of specific computational requirements, programming language / package specifics / versions is greatly appreciated and provides a good model of best practice.

- The Authors offer an expanded discussion on vaccine effectiveness, which is appreciated.

There is improved acknowledgment of computational cost for implementability of these forms of modeling approaches.

- Once again, critical to highlight the laudable commitment to open source code, the inclusion of local authors, and the explicit acknowledgement of the importance of local experts (found in the discussion). This is an excellent example of necessary good practice for research in the public health sphere in LMICs.

Unfortunately, following revision I have a few points of outstanding concern:

- The title is a bit odd, intuitively suggesting that the Authors compare models using subnational case data to those that do not, which is not what is presented.

- As stated above, the paper still fails to take a truly methodological focus. Discussion of methodological challenges in modeling in data frail environments is an important and underrepresented topic and there is great potential for this paper to highlight that issue. To do so, however, I would like to see the methodological challenges and solutions take focus as a result unto themselves rather than remaining in the methods description.

- It remains unclear if the models described are intended for moderate burden or high burden settings, and the definition of high burden is unclear; this issue is primarily seen in the introduction.

- In the introduction the Authors discuss important aspects of persistent susceptibility being attached to "pockets" and social networks but little discussion is given as to whether the model as proposed can truly identify this type of hyper heterogeneous susceptibility. A discussion of whether such dynamics can be captured as well as when and where complementary field based approaches to assess local heterogeneity in immunity could/would be needed would have been appreciated.

In the introduction and discussion, the Authors seem to claim that identification of subnational susceptibility is critical to effective targeted SIA policy. While identifying susceptible pockets is of course relevant, this statement feels perhaps too strong and they do not provide justification of why this type of approach is truly critical against simpler local solutions that look at coarser grained estimates of low vaccine coverage. It also ignores other aspects that go into targeted immunization campaigning, such as consideration of populations most at risk, resource constraints, and other local factors.

Particularly in the introduction the Authors conflate vaccine effectiveness with efficacy, which should be corrected.

- While the addition of acknowledging rubella misclassification is appreciated, this discussion is underdeveloped and the generalizability of the same model across settings of differential rubella burden is not justified. This remains particularly important in areas that continue to use mono antigen measles vaccine, which is true of several high burden settings.

- In my prior review I mentioned the need for more clarity on operational utility and model validation. While within the context of shifting the focus to methodological challenges specific operational recommendations for Ethiopia become less pertinent, the broader need for a discussion of validation and operational purpose remains relevant. It is important for the Authors to critically consider both how such a model can/should be used and whether it is justifiable for countries to be asked to buy into a model that would likely create not only financial burden but a dependence on (likely) global north countries for continued implementation. While the Authors note that this tool should be used not as a replacement for but rather in concert with simpler local tools, a deeper reflection on why such addition is necessary/warranted would have been greatly appreciated. It is worth noting here that a perfectly acceptable explanation is that such modeling efforts could be justified as "one off" validations assessing how well (or not) simpler methods work compared to more statistically complex approaches; but such interpretation should be made explicit.

- In particular, a more thoughtful discussion of how such a model could be validated, and the limitations of such validation remains critical. Ethically, if such models are meant to be operationalized for decision making, it is necessary to have a mechanism of validation. At present, the Authors do not fully address this issue. In the context of a methods based paper, this could be approached from a perspective of challenges in such validation; but it should still be meaningfully considered. If the conclusion is that such validation may be unrealistic in the context of application, a discussion of the ramifications of that conclusion is warranted.

I would like to underscore that despite the concerns sighted above, modeling work like this remains deeply relevant and, in particular, work like this that clearly describes the methodological difficulties and potential solutions that could be used in those efforts is a valuable contribution. These types of models are a point of active research and the absence of publications discussing more technical points and challenges of implementation only acts to limit the progress of the field.

**Have the authors made all data and (if applicable) computational code underlying the findings in their manuscript fully available?**

Reviewer #1: **No: ** Neither code nor data is available

Reviewer #2: **No: ** This form of local health data is not something that would normally be made public and the absence of disclosure should not be held against the authors

PLOS authors have the option to publish the peer review history of their article (what does this mean? ). If published, this will include your full peer review and any attached files.

**Do you want your identity to be public for this peer review?** For information about this choice, including consent withdrawal, please see our Privacy Policy .

Reviewer #1: No

Reviewer #2: No
---

## [Decision Letter · Decision Letter 2]

26 Feb 2025

Dear Ms. Sbarra,

We are pleased to inform you that your manuscript 'Fitting dynamic measles models to subnational case notification data from Ethiopia: methodological challenges and key considerations' has been provisionally accepted for publication in PLOS Computational Biology.

Best regards,

Yamir Moreno

Academic Editor

PLOS Computational Biology

Virginia Pitzer

Section Editor

PLOS Computational Biology

The reviewer has a few suggestions for improving the figures, but if you choose to address these, it can be done during the final proofing stage.

Reviewer's Responses to Questions

**Comments to the Authors:**

Reviewer #1: Minor outstanding comments are just a few suggestions to improve the figures:

Figures 3 and 5: Colour does not seem needed

Figure 3, 6, and 7: Font very small

Figure 7: Too small

The rest (earlier comments) seems to have been resolved acceptably.

**Have the authors made all data and (if applicable) computational code underlying the findings in their manuscript fully available?**

Reviewer #1: **No: **

PLOS authors have the option to publish the peer review history of their article (what does this mean? ). If published, this will include your full peer review and any attached files.

**Do you want your identity to be public for this peer review?** For information about this choice, including consent withdrawal, please see our Privacy Policy .

Reviewer #1: No

---

## [Editor Report · Acceptance letter]

PCOMPBIOL-D-24-00177R2

Fitting dynamic measles models to subnational case notification data from Ethiopia: methodological challenges and key considerations

Dear Dr Sbarra,

I am pleased to inform you that your manuscript has been formally accepted for publication in PLOS Computational Biology. Your manuscript is now with our production department and you will be notified of the publication date in due course.

With kind regards,

Anita Estes
